# Patterns of primates crop foraging and the impacts on incomes of smallholders across the mosaic agricultural landscape of Wolaita zone, southern Ethiopia

Yigrem Deneke[1,2]*, Aberham Megaze[1], Wondimagegnehu Tekalign[1], Taye Dobamo[1], Herwig Leirs[2]

1 Department of Biology, College of Natural and Computational Sciences, Wolaita Sodo University, Wolaita Sodo, Ethiopia, 2 Evolutionary Ecology Group, University of Antwerp, Antwerp, Belgium

* yigremk@gmail.com

**Data Availability Statement:** All relevant data are within the manuscript and its Supporting Information files.

## Abstract

Crop foraging by primates is a prevalent form of human-wildlife conflict, especially near protected areas. This behavior poses significant economic challenges for subsistence farmers, jeopardizing both livelihoods and conservation efforts. This study aimed to assess patterns of primate crop-foraging events and estimate maize damage in protected and unprotected fields in southern Ethiopia. Data were collected over 12 months between 2020 and 2021 in the Sodo Zuriya and Damot Gale districts of Southern Ethiopia. A team of six field experts and 25 farmers participated in the study, during which maize damage inflicted by primates was assessed using 25 deployed camera traps. Linear mixed models were used to explore the relationship between maize damage by primates and spatio-temporal variables. Olive baboons and grivet monkeys were found to target maize more frequently during June, July, and August. Olive baboons forage in the morning, while grivet monkeys do so in the afternoon. The average maize yield losses due to primate damage were 43.1% in protected fields and 31.4% in unprotected fields. Of the total damage, 43.1% occurred in protected fields situated 50 meters from the forest edge. Conversely, unprotected fields experienced lower rates of damage: 14.4%, 13.2%, 3.7%, and 0.1% at distances of 50 m, 100 m, 200 m, and 300 m from the forest edge, respectively. Camera traps captured 47 photos of baboons, 21 photos of grivet monkeys, and documented eight primate crop-foraging events. This study revealed that maize fields within 50 meters of the forest edge faced significant damage. Despite the use of wire mesh fencing, it was largely ineffective in deterring olive baboons and grivet monkeys. Additionally, while human guarding is often considered an effective protective strategy, these findings suggest its ineffectiveness due to inconsistent implementation. Overall, this study provides valuable insights for promoting primate conservation and mitigating human-primate conflicts.

**Funding:** This work was funded by the Flemish Interuniversity Council (VLIR-UOS, ET2019TEA485A102). The funders had no role in study design, data collection, and analysis, decision to publish, or preparation of the manuscript.

**Competing interests:** The authors have declared that no competing interests exist.

## Introduction

Crop foraging occurs when wild animals leave their natural habitats to pilfer crops cultivated by farmers for household consumption [1, 2]. This issue has persisted since humans and wild animals began sharing landscapes and resources. In protected areas, human-wildlife conflict is severe and presents a growing challenge, mainly due to mismatches between conservation interests and the improvement of local residents' livelihoods [3, 4]. The frequency of crop foraging and the resulting damage may vary along a distance gradient from natural habitats to human-modified landscapes [5, 6]. A commonly reported pattern is that wild animals move from uncultivated habitats to damage crops [7, 8]. Crops grown near forest edges are generally more susceptible to damage than those grown farther away from forests [4, 9–13]. Moreover, the intensity of crop foraging largely depends on the type of foraging species, the crop species grown, and the season among others [14].

Finding effective ways to resolve frequent conflicts between people and wildlife is essential for fostering coexistence outside protected areas. Identifying successful methods will significantly enhance conflict resolution and wildlife conservation [4]. Current threats to wildlife arising from such conflicts require strategies to manage and mitigate them for populations to persist and thrive [15]. Conflict resolution is also crucial in reducing the vulnerability of people affected by wildlife, by minimizing the extent of damage sustained [16]. However, the success or failure of any mitigation technique is likely to be site- and species-specific, requiring appropriate and site-specific actions. Such actions depend on factors such as the species, location, timing, and the historical and socio-ecological context [5, 17]. For example, the activity patterns and ranging behavior of different species, which influence daily and seasonal damage patterns and determine the types of crops targeted, can significantly impact the effectiveness of mitigation strategies [17].

Mammals such as baboons, monkeys, bush pigs, porcupines, and elephants are recognized as some of the most destructive crop foragers across various regions of Africa [13, 18–21]. These mammals significantly impact agricultural production by causing damage to cereals, root crops, and fruits through mechanisms such as feeding and trampling, which in turn adversely affect crop yields and household incomes [5, 22]. Among the various crops foraged by primates, maize (*Zea mays*) was selected for this study due to its status as a major staple cereal crop that supports the livelihoods of millions of smallholders in Ethiopia [23]. Similarly, maize is the most dominant staple crop in Wolaita Zone in terms of production, occupying 42% of the land covered by grain crops [24]. It serves as a primary food source in many African countries, providing both protein and energy [25]. Consequently, primates, especially monkeys, show a strong preference for maize; once they have tasted it, they seem to highly value it, which explains their frequent forage on maize fields [26]. Primates that forage on subsistence farmers' crops are of particular concern, as they threaten their livelihoods [18–20]. Human-primate conflict has been widely studied across several African countries; including Guinea-Bissau [27, 28], Madagascar [29], Rwanda [30], South Africa [31], Tanzania [32], Ethiopia [33, 34], and Uganda [35–37]. Primates are frequently identified as the most common crop foragers, particularly targeting maize crops in tropical regions of Latin America [38]. Farmers in Bengo, Indonesia, have also reported primate-induced damage to maize crops through feeding [39]. In the Kavrepalanchok District of Nepal, maize is similarly recognized as a key crop affected by the foraging activities of primate macaques [40]. A study in the Budongo Forest Reserve, Uganda, found that baboons consistently focused their foraging activities on maize throughout the year, even when other crops were available [41]. Additionally, they imposed indirect costs, such as the labor needed to protect the crops [41]. Another similar study in the same area confirmed that primates were the primary foragers of maize crops [42].

Primate maize crop foraging has also been reported in the southwestern region of Mole National Park in Ghana, which is known for its diverse species of primates [43]. In West Africa, maize has been identified as the crop most frequently consumed by primates [19]. A study conducted in the forest-agricultural landscape mosaic of Taita Hills, Kenya, also found that maize to be the most frequently attacked crop by primates [44]. In the Serengeti National Park, Tanzania, maize was identified as the crop most commonly damaged by baboons and other wild animals [45]. These studies recognize the seriousness of human-primate conflict and its drastic impact on the livelihoods of rural households. Subsistence farmers, who heavily rely on their agricultural production, face a serious threat to food security due to wildlife crop foraging, especially by primates. Additionally, the livelihoods of local communities near protected areas largely depend on agriculture, which is highly vulnerable to crop foraging [37, 44]. In Ethiopia, various wild animals, including both small and large mammals, have been reported to forage crops [18]. In the southwestern part of the country, several large mammals such as olive baboons, bush pigs, vervet monkeys, porcupines, and warthogs have been identified as significant crop foragers [6, 46]. A study in southwest Ethiopia found that maize was one of the most vulnerable crops to foraging by olive baboons and grivet monkeys [47]. Similarly, in southern Ethiopia, interviewed farmers reported that primates were the most frequent crop foragers, causing substantial damage to maize crops [48]. However, the frequency and extent of crop raiding incidents may vary along a distance gradient from wildlife habitats [5, 6]. Moreover, the consequence of such incidence on crops has varying impacts on the income of smallholder farmers across mosaic agricultural landscapes. Despite this variation, little is understood about the pattern and socio-economic impacts of crop foraging by primates in the biodiversity hotspots of Southern Ethiopia, Sodo Zuriya and Damot Gale districts.

The focus of this study was to comparatively assess the patterns of crop foraging by primates, the extent of maize damage, and its impact on the income of smallholder farmers in both protected and unprotected fields at varying distances (50m, 100m, 200m, and 300m) from the forest edges. In contrast to previous studies that emphasized farmers' perceptions of human-primate interactions during crop foraging events in unprotected fields, this research involved direct monitoring. Consequently, a participatory approach was employed, with maize damage assessed through collaboration among field experts, farmers, and researchers. Camera-trapping techniques were used to monitor crop foraging patterns and quantify the extent of maize damage caused by primates, allowing for a comparative analysis. This study hypothesized that the extent of maize damage could be analyzed by modeling crop foraging events using linear mixed modeling (LMM), taking into account variables such as the distance of fields from the forest, the duration of foraging events, and crop phenology. Moreover, this study compared protected maize fields, safeguarded with wire mesh, human guards, scarecrows, and thorny bushes, with unprotected maize fields. The effectiveness of these protective measures was further evaluated with the goal of developing improved mitigation strategies and promoting primate conservation in the forest-agricultural mosaic of the Wolaita Damota Areas.

## Materials and methods

### Study area

The study was conducted in the Sodo Zuriya and Damot Gale districts, located approximately at 6.54˚N 37.45˚E through 6.9˚N 37.75˚E in the Highlands of Southern Ethiopia. The study sites included the Gurumu Woyde, Kokate Marachere, Konasa Pulasa, Damot Waja, and Dalbo Wogene sub-districts (S1 Fig). The study area covers 380 km$^2$ and is primarily situated atop Mt. Damota. The Damota Community Managed Forest was established in January 2006

through collaboration between the Sodo community and World Vision Ethiopia. The aim was to restore and protect the montane high forest on the slopes of Mount Damota. The land is collectively owned by five Sodo Zuriya and Damot Gale Communities, who secured the site and obtained land user-rights certificates from the Ethiopian Government in 2006. Furthermore, the Ethiopian government has supported the community's ownership of carbon rights trading, allowing them to earn revenue from carbon offsets [49]. Additionally, cooperatives were established to manage the protected areas and provide education to the local community on mitigating crop damage caused by wildlife, thereby helping them maintain their livelihoods. According to the institute's assessment, the area also plays a role in global climate regulation [49]. This region experiences a dry period from October to March and a wet season from April to September, receiving 1450 to 1800 mm of rainfall, respectively [49]. The maximum rainfall occurs between June and September, with shorter rains falling in March and April [48]. The temperature ranges from 16°C to 24°C between the wet and dry seasons. The soil nutrients in the Damota area are suitable for growing maize [50].

The Damota Community Managed Forest is characterized by rugged topography and diverse agro-ecology, fauna, and flora. The Damota area is characterized by Dega and Woina Dega zones, with altitudes ranging from 1,480 to 2,855 meters above sea level [51]. The vegetation is marked by various types, including evergreen needle-leaved, deciduous needle-leaved, evergreen broadleaved, and deciduous broadleaved forests, mixed with shrubland, herbaceous vegetation, herbaceous wetland, moss and lichen, sparse/bare vegetation, and cropland [49]. Dominant plant species in this area include woodland waterberry (*Syzygium guineense*), African juniper (*Juniperus procera*), Broad-Leaved Croton (*Croton macrostachyus*), briar root (*Erica arborea*), common olive (*Olea europaea*), and Shittim Wood (*Acacia hockii*), [49]. These vegetation and plant species provide food and serve as suitable habitats for mammals, particularly primates. The region is home to various large and medium-sized mammals, such as olive baboons (*Papio anubis*), grivet monkeys (*Chlorocebus aethiops*), duikers (*Sylvicapra grimmia*), common bushbucks (*Tragelaphus scriptus*), Guenther's dikdik (*Madoqua guentheri*), and porcupines (*Hystrix cristata*). Golden jackals (*Canis aureus*), black-backed jackals (*Canis mesomelas*), leopards (*Panthera pardus*), African civets (*Civettictis civetta*), and spotted hyenas (*Crocuta crocuta*) [49]. The entire area sustains a population of 16,342 people [52]. In Mount Damota, farmers typically possess very small plots of land. The range of landholding sizes spans from 0.06 to 1.75 hectares, with an average size of 0.5 hectares [53]. The Wolaita zone, characterized by a highland perennial farming system, supports a diverse array of crops [54]. Primary food crops in this region include maize, teff, various vegetables, and root and tuber species such as cassava, yam, potato, sweet potato, and taro [54]. Additionally, tropical and temperate fruit tree crops like banana, avocado, mango, and apple are cultivated in the Wolaita Areas [54].

## Experimental setup

The experimental setup was established using 25 maize fields. Maize fields in these areas tend to be quite small, often measuring around 10m x10 m, and are interspersed with fields growing different crops. For the purposes of this study, maize fields were selected to assess the extent of damage caused by primates. Ten maize study plots were situated 50 meters from the forest edge and they were used to compare protective measures in the villages of Gurumu Woide and Kokate Marachare. The protected study plots were safeguarded using wire mesh, human guardians, scarecrows, and thorny bushes, while the unprotected fields remained open/control (S2 Fig). Two farmers in the area were hired as field guards to protect two maize fields, working seven days a week from dawn to dusk throughout the six-month maize harvest in 2020 and

2021. These farmers chase, shout at, and sometimes throw stones at wildlife entering the maize fields. Furthermore, a total of 15 unprotected maize study plots were set up (S1 Table), including Gurumu Woide, Kokate Marachare, Delbo Wogene, Damot Waja, and Konasa Pulasa. The study plots were located at varying distances: 100 meters, 200 meters, and 300 meters from the forest edge. Maize damage assessments were compared at varying distances by evaluating an open maize field located 50 meters from the forest edge, along with individual fields situated 100 meters, 200 meters, and 300 meters away from the forest edge. The distances of each study plot farthest from the forest edge were measured using a Garmin 72H Global Positioning System (GPS) device. Distances from field edges to reference features or structures (e.g. trees, paths, or huts) were recorded to aid in distance estimation (S3 Fig).

A study plot measuring 10m x 10m was designated in each study field for this research (S1 Table). Within these study plots, the high-yielding maize variety BH-546, which is well-suited for the region's agro-ecology, was sown. Maize seeds were sown early in the rainy season, typically in April, reaching the milky stage in late July and ripening by mid-August, with harvesting in September. Prior to sowing, oxen-drawn ploughs were used to prepare the fields by creating rows. Initially, 580 seeds were sown in each study plot in both the 2020 and 2021 maize cropping seasons. However, in one field (Field No. 25) seeds were removed or added by the farmer, resulting in 532 seeds (19 rows x 28 seeds) during the 2020 maize cropping season and 627 seeds (19 rows x 33 seeds) during the 2021 maize cropping season. Each hole received one seed, with a planting distance of 40 cm x 30 cm, while maintaining a distance of at least 50 meters between one maize study plot and the next. All cultivation practices, including fertilizer application, planting, and weeding, were implemented in the maize fields. However, uneven germination of the sown maize seeds resulted in varying harvests across different plots. In this study, data were collected using (1) field experts and (2) camera traps.

## Field experts

Data on crop foraging events (CFE) by primates were collected by six field experts, five of whom are agriculture and rural development office workers, and one is a village administrator. These experts were trained by researchers to ensure a thorough understanding of the subject. Each field expert was expected to monitor and assess the CFE in both olive baboons and grivet monkeys. They actively participated in the project during two maize harvest seasons (from April to August in both 2020 and 2021). Additionally, these experts were engaged in close collaboration with twenty-five local farmers during field observations and reporting. The overall data collection process was supervised by four researchers.

Researchers defined a primate crop foraging event (CFE) as occurring when one or more individuals of a species enter a field (i.e., cross a field boundary), trample or raid the crops, interact with one or more maize stems, and consume parts of the stems before leaving. The CFE begins when the first primate enters the field to feed on the maize stems and ends when the last primate leaves the field. The duration of the event was measured in seconds using a digital stopwatch. Primate age categories are defined as follows: adult (full species-sex-specific size), sub-adult (not fully grown, beyond infant development, and frequently exhibits independent behavior), and infant (developmentally small and dependent, often carried and maintaining close proximity to adults) [37]. Similarly, the extent of maize damage caused by primates was assessed based on crop phenology, focusing on the seedling, fruiting, and maturity stages. The seedling stages of maize (*Zea mays*) begin with the emergence of the first leaves (V1) and continue until the plant has developed around 5 to 18 leaves, culminating in the VT (Vegetative Tassel) stage. This stage occurs approximately two weeks before the flowering phase (R1), signaling the plant's transition to reproduction. The fruiting stages encompass the

reproductive phase of maize, beginning with pollination (R1 stage) and continuing through various stages of kernel development (R1 to R5) until the kernels develop a dent (R5). This marks the transition toward physiological maturity (R6). The maturity stage is reached at R6 when a black layer forms at the base of the kernel, signaling the cessation of water and nutrient flow, which indicates that the maize has achieved full grain maturity [55].

Field experts responded to the following questions: (1) What is the extent of primate damage to maize in protected versus unprotected fields? (2) When and during which months do primates forage on maize crops? (3) How long do primates typically remain during maize foraging events? (4) How frequently and at what times, do farmers report primate incursions? (5) Which crop-feeding species do farmers most commonly encounter? (6) What is the extent of primate induced maize damage in fields located at varying distances? (7) How many individual primates foraged maize and entered fields? (8) In what proportion do multiple and single primate forage events occur? (9) How many individual primates typically visit maize fields? (10) In which age categories are maize crop-raiding primates most commonly found? (11) To what extent is the income of smallholders affected by primate maize damage across mosaic landscapes?

Data were also collected regarding the presence or absence of humans on fields, the nature of on-field human activity, the extent of guarding behavior, and responses to crop-foraging primates. Crop damage was quantified by counting stems damaged by primates. Trained field experts assessed and recorded the damage caused by primates to maize daily at 18:00 hours.

## Camera traps

To gather information on the timing, frequency, and location of the crop foraging behavior of olive baboons and grivet monkeys within the 25 study plots, 25 Bushnell detection cameras (Browning Trail Camera, Model No. BTC-6HDX) were utilized in this study. These motion-trigger cameras were configured to capture and store data, including the date, time, location, and temperature for each photo. The cameras were set to take only one photo per trigger, with a 2-second interval between triggers [39]. Cameras were securely housed and locked in metal cases. A potential CFE was recorded when one or more individuals of olive baboons and grivet monkeys were merely present in the field [39]. An actual CFE was documented if the photo or video indicated physical manipulation and/or consumption of crop items [39, 56]. An interval of more than an hour between captured images was considered an independent CFE [39]. During this study, different camera traps were installed and dismantled on different days, resulting in varying numbers of trap days for each unit.

Cameras were installed in each study plot to monitor crop-foraging behavior. In this study, 30mm x 30mm stainless steel wire mesh with a wire diameter of 1.6 mm and a height of 2.5 meters was used. Each camera was equipped with 16GB or 32GB Class 4 SDHC memory cards for data storage. The camera traps were monitored by farmers to prevent theft. Data from the camera traps were collected from April to September in both 2020 and 2021, with cameras installed in each of the 25 maize fields for four consecutive trapping days. The cameras operated for a total of 192 trapping days. During camera installation, the following information was collected: camera ID, GPS position, date, and altitude. Subsequently, photos and videos from the camera traps were downloaded onto a laptop. Each photo and video was checked for the presence of wildlife and other relevant information. The presence of humans and dogs, among other factors, was investigated. Photos containing baboons and monkeys that could damage the crops were numbered and placed in a digital folder. All saved photos and videos were catalogued, and the associated information was recorded in a spreadsheet.

## Data analysis

Data were analyzed using SPSS Version 27 for Windows (SPSS Inc., Chicago, USA). Tests were two-tailed, and results were deemed statistically significant when p ≤ 0.05. The images captured by camera traps were interpreted to determine the frequency and timing of crop foraging events. Descriptive statistics were employed to analyze crop foraging data. A chi-square test was conducted to examine the variation in maize damage by primates across different variables, including primate species raiding duration, multiple versus single raid events, primate CFE timing, and age-category of raiding in single or group. Mann-Whitney U test was used to compare the raiding durations of primate CREs among different age categories of primate species. The Spearman correlation coefficient assessed the relationship between the number of individuals entering a field and the number at the forest edge prior to raiding. The independent sample t-test compared estimates of maize damage among variables such as the number of individuals raiding, Primate CREs, farm distance, duration of raiding, and crop phenology. One-way ANOVA and the F-test were employed to compare estimates of maize damage between preventive and non-preventive strategies during the cropping seasons, as well as between single and multiple raids. The extent of primate assaults on maize in preventive and non-preventive maize fields during different crop phenological stages was analyzed using R version 4.4.1 (bplot function in the Rlab package) [57]. Linear mixed models (LMMs) were used to analyze various spatial-temporal variables, including fixed factors (distance, duration, and phenology) and random factors (primate CREs and the number of individuals raiding). In LMMs, it is typically assumed that the data model distribution is normally distributed. The link function used was the identity link, which means that the expected value of the response variable is modeled directly as a linear combination of the fixed and random effects. The response variable was the rate of maize damage, and the analysis was conducted using R version 4.4.1 [57]. Maize damage was reported in three ways: the average number of maize stems or cobs affected, the estimated amount of maize damaged in kilograms, and the proportion of maize damage caused by primates relative to the expected harvest. To calculate monetary loss, the market price of maize per kilogram was converted to US dollars using the prevailing exchange rate at the time of the survey. Additionally, it was estimated that the seeds from a single maize stalk weighed approximately 0.2 kg, yielding around 1.5 ears (or cobs) after harvest.

## Ethics statement

The study was approved by the institutional ethics committee, adhering to the established ethical guidelines of Wolaita Sodo University, under Reference number. WSU15/12/915. Subsequently, permission was obtained from the Wolaita Zone Agriculture, Environment, Forest, and Climate Change Regulatory Office, as well as the respective district authorities. Verbal consent was obtained from each study participant. All social data of the study participants were kept confidential and anonymized before analysis. In addition, there was no direct interaction between field personnel and the subjects (the primates) in such a way as to harm the animals or interfere with their freedom in nature, such as by way of capture or trapping.

## Results

### Farmer-reported crop foraging species and crop damage assessments in protected and open or control fields

Twenty-five farmers consistently reported that olive baboons, porcupines, and grivet monkeys were the primary culprits responsible for the most severe crop damage to maize, exhibiting a high frequency of crop foraging events. Additionally, some farmers (N = 10) suggested that

bushbuck might also be involved in crop foraging. However, the reported frequency of crop foraging events for bushbuck in maize fields was notably low, occurring only 24 times (Table 1). The average percentage of maize cobs lost by olive baboons in wire mesh, human guard, scarecrow, and thorny bush setups was 8.23% (equivalent to 72.8 maize stems/cobs), 7.38% (65.3 maize stems/cobs), 9.82% (86.8 maize stems/cobs), and 9.45% (83.5 maize stems/cobs), respectively, at 50 meters from the forest edge (S2 Table). In two open/control fields, the average percentage of maize cobs lost to olive baboons was 10.04% (88.8 maize cobs) at 50 meters. In unprotected fields, the average percentage of maize cobs lost to olive baboons was 1.53% (13.5 maize cobs) at 100 meters, 0.4% (3.6 maize cobs) at 200 meters, and 0.1% (0.9 maize cobs) at 300 meters (S2 Table). For grivet monkeys, the average percentage of maize cobs lost in fields with wire mesh, human guards, scarecrows, and thorny bushes was 0%, 1.83% (6.3 maize cobs), 3.8% (13 maize cobs), and 2.63% (9 maize cobs), respectively, with these fields also located at 50 meters. In two open/control fields, the average percentage of maize cobs lost to grivet monkeys was 4.38% (15 maize cobs) at 50 meters. In unprotected fields, the average percentage of maize cobs lost to grivet monkeys was 11.65% (39.9 maize cobs) at 100 meters, 3.3% (11.3 maize cobs) at 200 meters, and 0% at 300 meters from the forest edge (S2 Table). Overall, the average percentage of maize cobs lost to these two primate species in protected and two open/control fields was 43.14% (336.7 maize cobs) and 14.42% (103.8 maize cobs), respectively, at 50 meters. In unprotected fields, the average percentage of maize cobs lost to these two primate species was 13.18% (53.4 maize cobs) at 100 meters, 3.7% (14.9 maize cobs) at 200 meters, and 0.1% (0.9 maize cobs) at 300 meters, respectively (S2 Table). The resulting average monetary losses for farmer households amounted to 1,103 ETB (equivalent to 32 US dollars) across the twenty-five maize fields (S2 Table).

Farmers reported that the average percentage of maize damaged by olive baboons at both the Gurumu Woide and Kokate Marachare study sites was 23.6% in fields with wire mesh, 21.0% with a human guard, 28.2% with a scarecrow, and 27.2% in thorny bush fields (Fig 1 and S1 File). The results of a one-way ANOVA indicated that damage in maize fields was significantly higher in thorny bush fields compared to damage levels in fields with wire mesh, human guards, and scarecrows ($F_{(2,9)}$ = 292.5, $p < 0.001$).

Farmers reported that the average percentage of maize damaged by grivet monkeys at the Kokate Marachare study site was 0% in fields with wire mesh, 24.1% with a human guard, 44.8% with a scarecrow, and 31.0% in thorny bush fields (Fig 2 and S2 File). The results of a one-way ANOVA indicated that the damage in maize fields was significantly higher in thorny bush fields compared to damage levels in fields with wire mesh, human guards, and scarecrows ($F_{(2,9)}$ = 5.4, $p < 0.05$).

## Camera trap results

The cameras recorded 47 photographs of baboons and 21 photographs of grivet monkeys (S2 Table). Of the 47 photographs of baboons, only 3 were confirmed as actual CFE, while the remaining 44 were potential CRE. Similarly, out of the 21 photographs of grivet monkeys, only

**Table 1. Farmer responses on crop-foraging species from April to September in 2020 and 2021 showed the following involvement: Bushbuck (n = 10), grivet monkeys (n = 17), olive baboons (n = 22), and porcupines (n = 25).**

| Pest species | Number of farmers reporting the species | Frequency of CFE |
|---|---|---|
| Baboon (*Papio Anubis*) | 22 | 80 |
| Grivet monkey (*Chlorocebus aethiops*) | 17 | 45 |
| Porcupine (*Hystrix*) *cristata*) | 25 | 75 |
| Common bushbuck (*Tragelaphus scriptus*) | 10 | 24 |

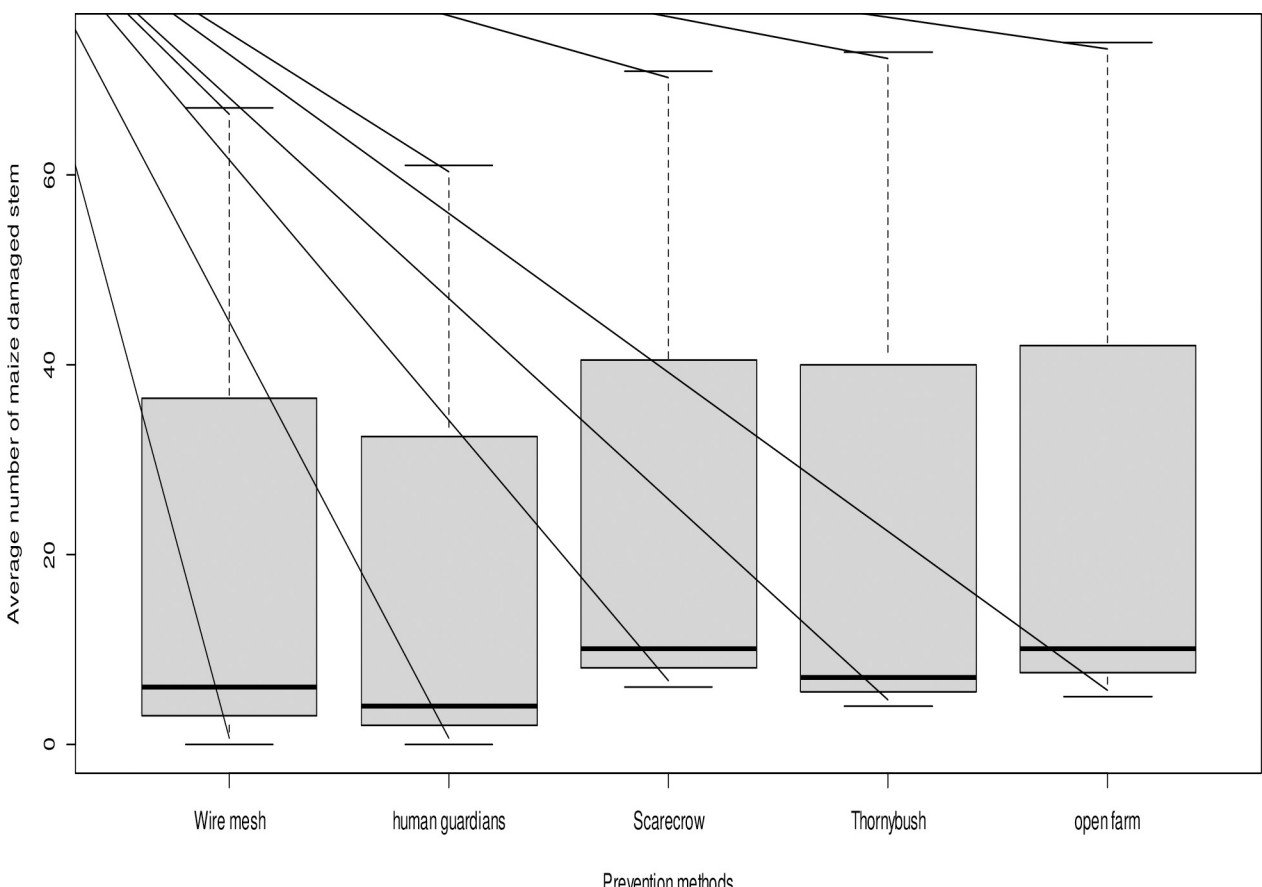

**Fig 1. The average of maize stems (≈number of cobs) damaged within 10m x 10m study plots by olive baboons was examined in relation to various preventive methods at a distance of 50 meters from the forest edge during the 2020 and 2021 maize cropping seasons and crop phenology in the Gurumu Woide and Kokate Marachare (GW) sub-district.** The boxplot illustrates a significant difference in crop damage among different prevention methods (p < .001).

2 were confirmed as actual CFEs, with the remaining 19 being potential CREs. Notably, the longest CRE event, recorded by camera IDs A3 and E1, occurred in scarecrow and open maize fields (Table 2, S4 Fig, S3 Table and S1 video).

## Determinants of maize damage: Field distance, duration, phenology, and timing of crop foraging events

In this study, the spatial-temporal variables affecting maize damage by primates were analyzed using a linear mixed model. The model indicated that farms located 200 meters from the forest edge experienced significantly fewer maize foraging incidents compared to farms located 50 meters from the forest edge (LMM: t = -2.728, DF = 256.9, p < 0.007). The duration of maize foraging incidents was significantly longer, lasting 6.1–9 minutes, compared to durations of 0.1–3 minutes (LMM: t = -1.993, DF = 182.9, p < 0.04). Similarly, maize foraging incidents were significantly higher during both the fruiting stage (LMM: t = -11.656, DF = 98.9, p < 2e-16) and the maturity stage (LMM: t = -13.53, DF = 176.05, p < 2e-16) compared to the seedling stage (Table 3 and S4 Table).

The median raid duration ranged from 15.1 to 18 minutes, with a mean of 3.78 and a standard deviation of 0.66 for primates (S5 Fig). Raid durations were significantly shorter when

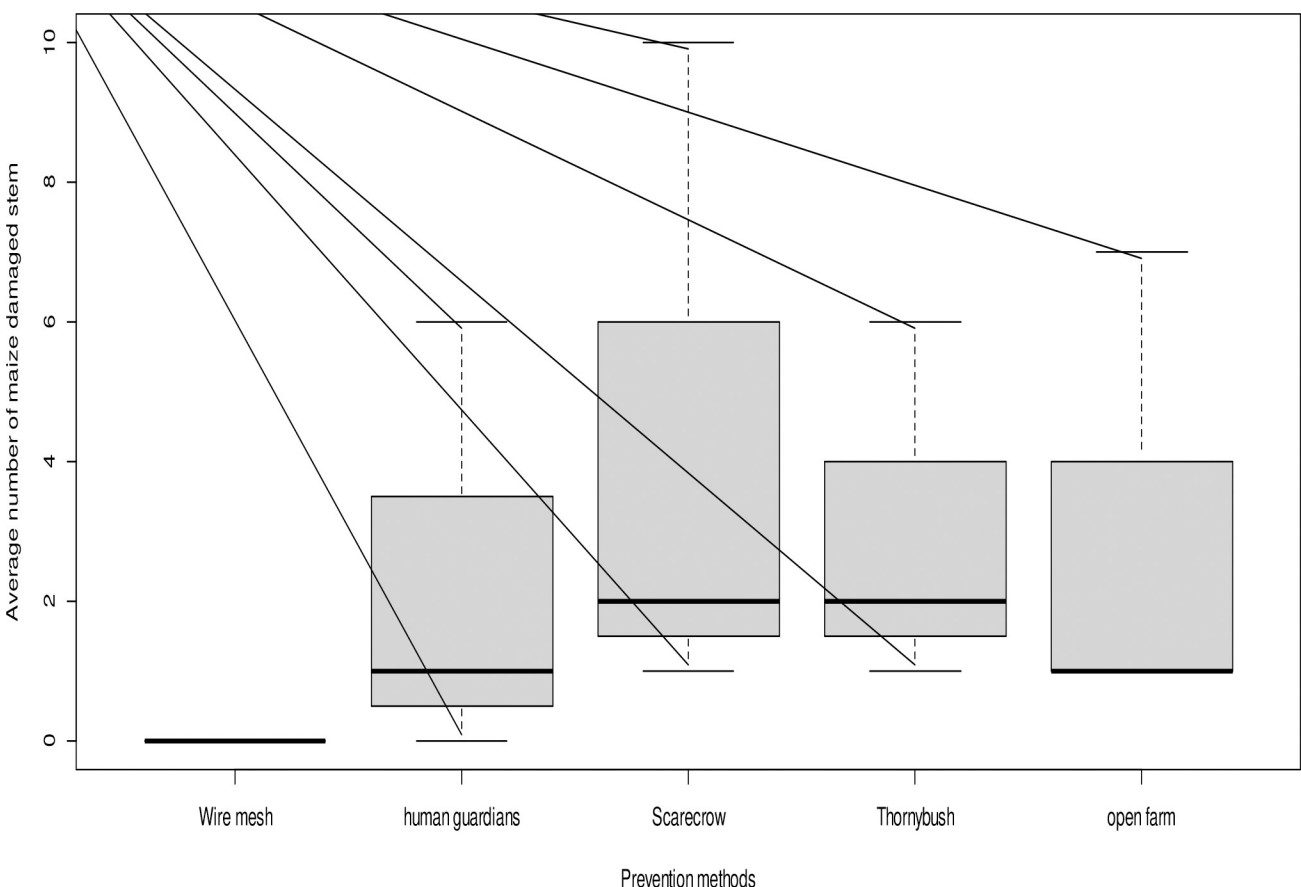

**Fig 2. The average of maize stems (≈the number of cobs) damaged within 10m x 10m study plots by grivet monkeys illustrates the relationship with various prevention methods at a distance of 50 meters from the forest edge during the 2020 and 2021 maize cropping seasons and crop phenology in the Kokate Marachare (KM) sub-district.** The boxplot shows a significant difference in crop damage with different prevention methods (p < .005).

carried out by single individuals (median 1 minute, SD = 0.42) compared to raids by two or more individuals (median 3 minutes, SD = 2.42), as confirmed by the Mann-Whitney U test (n (single) = n (two+) = 38, U = 34.0, p < 0.001). The majority of CREs, approximately 70%, lasted between 0.1 and 12 minutes (S5 Fig).

According to responses from twenty-five farmers, a higher frequency of maize cobs was reported to be plucked by primates in July, with 524 ± 3.8 cobs in 2020 and 539 ± 4.6 cobs in 2021. Moderate frequencies of maize cobs were reported to be plucked by primates in June and August, with 216 ± 4.6 and 64 ± 2.1 cobs in 2020, and 240 ± 5.2 and 25 ± 1.6 cobs in 2021, respectively. The lowest frequencies of maize cobs were reported to be plucked by primates in April and May for both 2020 and 2021 (Fig 3). Farmers observed that baboons typically fed on crops early in the morning, while grivet monkeys fed on crops throughout the day. According to farmers, neither baboons nor grivet monkeys were seen eating on crops at night. Baboon crop feeding events (CFEs) occurred throughout the day but not in a uniform distribution, as revealed by photographic data from five locations (Chi-square goodness of fit: $\chi^2$ = 32.36, df = 12, p < 0.001). Similarly, grivet monkey CFEs occurred throughout the day, also with a non-uniform distribution, based on photographic data from five locations (Chi-square goodness of fit: $\chi^2$ = 35.86, df = 8, p < 0.001). Morning CFEs were more common in baboons (6:00–7:00 a.m.) than afternoon CFEs (2:00–3:30 p.m.). In contrast, CFEs were more common

**Table 2. Camera trap data from 25 maize fields during 2020 and 2021 captured images of olive baboons (n = 47) and grivet monkeys (n = 21).** Of the baboons, 3 images were CFEs and 44 were CREs. For grivet monkeys, 2 images were CFEs and 19 were CREs.

| Study sites | Camera ID | Distance to forest edge | Preventive and Non-preventive measures | Olive baboon | | Grivet monkey | |
|---|---|---|---|---|---|---|---|
| | | | | CRE | CFE | CRE | CFE |
| Gurumu Woide | A1 | 50m | Wire mesh | 4 | 0 | 0 | 0 |
| | A2 | 50m | Human guard | 10 | 0 | 0 | 0 |
| | A3 | 50m | Scarecrow | 12 | 3 | 0 | 0 |
| | A4 | 50m | Thorny bushy | 6 | 0 | 0 | 0 |
| | A5 | 50m | Open/control | 9 | 0 | 0 | 0 |
| | A6 | 100m | Open | 3 | 0 | 0 | 0 |
| | A7 | 200m | Open | 0 | 0 | 0 | 0 |
| | A8 | 300m | Open | 0 | 0 | 0 | 0 |
| Kokate Marachare | B1 | 50m | Wire mesh | 0 | 0 | 0 | 0 |
| | B2 | 50m | Scarecrow | 0 | 0 | 1 | 0 |
| | B3 | 50m | Thorny bush | 0 | 0 | 1 | 0 |
| | B4 | 50m | Open/control | 0 | 0 | 1 | 0 |
| | B5 | 50m | Human guard | 0 | 0 | 0 | 0 |
| | B6 | 100m | Open | 0 | 0 | 1 | 0 |
| | B7 | 200m | Open | 0 | 0 | 0 | 0 |
| | B8 | 300m | Open | 0 | 0 | 0 | 0 |
| Delbo Wogene | C1 | 100m | Open | 0 | 0 | 1 | 0 |
| | C2 | 200m | Open | 0 | 0 | 0 | 0 |
| | C3 | 300m | Open | 0 | 0 | 0 | 0 |
| Damot Waja | D1 | 100m | Open | 0 | 0 | 1 | 0 |
| | D2 | 200m | Open | 0 | 0 | 0 | 0 |
| | D3 | 300m | Open | 0 | 0 | 0 | 0 |
| Konasa Pulasa | E1 | 100m | Open | 0 | 0 | 11 | 2 |
| | E2 | 200m | Open | 0 | 0 | 2 | 0 |
| | E3 | 300m | Open | 0 | 0 | 0 | 0 |
| Total | | | | 44 | 3 | 19 | 2 |

in the early afternoon (11:00 a.m.–12:00 p.m.) for grivet monkeys than in the morning (6:00–7:00 a.m.) during both 2020 and 2021 years. Farmers reported no baboon CFEs in all five locations between 11:00 a.m. and 6:00 p.m. during both 2020 and 2021 years (Fig 4).

## Primate crop raiding events, field visits, and age category composition of crop-raiding primates

A total of 367 primates were observed at the forest edges immediately before or during crop raiding events (CREs). Out of these, 367 individuals, accounting for 75%, ventured into fields (Table 4). Among 95 crop raiders, 75 CREs were attributed to olive baboons (79%), while 20 CREs were attributed to grivet monkeys (21%). Notably, olive baboons were significantly more likely to be found near the forest edge than grivet monkeys, as indicated by the Kruskal-Wallis test ($\chi^2$ = 263.1, df = 1, p < 0.001). The number of individuals entering a field showed a positive correlation with the number at the forest edge prior to raiding, which was confirmed by the Spearman's rank correlation coefficient (rs = 0.434, n = 95, p = 0.006). This correlation persisted even when humans were present in the field, with a Spearman's rank correlation coefficient of rs = 0.324, n = 59, and p = 0.04. Regarding the composition of CREs, the majority (36.1%) involved three or fewer individuals, while 47.8% consisted of a single individual or a

**Table 3. A linear mixed model (LMM) analyzed maize damage caused by primates during CREs (n = 95) with the following significance results: Distance (P < 0.007), duration (P < 0.04), fruiting stage (P < 2e-16), and maturity stage (P < 2e-16).**

| Parameters | Estimate | Std. Error | DF | t value | Pr (>|t|) |
|---|---|---|---|---|---|
| (Intercept) | 66.646 | 4.424 | 30.611 | 15.064 | 1.06e-15 *** |
| distance_farm100m | 1.848 | 2.004 | 256.286 | -0.922 | 0.357 |
| distance_farm200m | -10.088 | 3.698 | 256.976 | -2.728 | 0.007 ** |
| distance_farm300m | -6.388 | 4.196 | 257.913 | -1.523 | 0.129 |
| duration of_raiding3.1–6 minute | -3.276 | 2.312 | 257.931 | -1.417 | 0.158 |
| duration of_raiding6.1–9 minute | -6.466 | 3.244 | 182.907 | -1.993 | 0.048 * |
| duration of_raiding9.1–12 minute | -3.517 | 4.119 | 217.458 | -0.854 | 0.394 |
| duration of_raiding12.1–15 minute | -7.025 | 5.300 | 147.578 | -1.325 | 0.187 |
| duration of_raiding15.1–18 minute | -9.031 | 5.434 | 218.392 | -1.662 | 0.098 |
| duration of_raiding18.1–21 minute | -6.752 | 6.370 | 232.020 | -1.060 | 0.290 |
| duration of_raiding21.1–24 minute | -8.664 | 6.813 | 248.224 | -1.272 | 0.205 |
| duration of_raiding24.1–27 minute | -11.756 | 7.637 | 245.037 | -1.539 | 0.125 |
| duration of_raiding27.1–30 minute | -11.639 | 8.685 | 228.281 | -1.340 | 0.182 |
| duration of_raiding>30 minute | -8.555 | 10.282 | 227.031 | -0.832 | 0.406 |
| crop_phenology_fruiting | -46.620 | 3.999 | 98.983 | -11.656 | < 2e-16 *** |
| crop_phenology_maturity | -55.256 | 4.084 | 176.050 | -13.530 | < 2e-16 *** |

Significance codes: 0 '***' 0.001 '**' 0.01 '*' 0.05 '.' 0.1 ' ' 1

pair. Only 16.1% of CREs involved more than five individuals (S6 Fig). It's worth noting that baboons raided in significantly larger groups than grivet monkeys. Most olive baboon raiding groups, comprising fewer than five individuals, accounted for 78% of the total raids. In

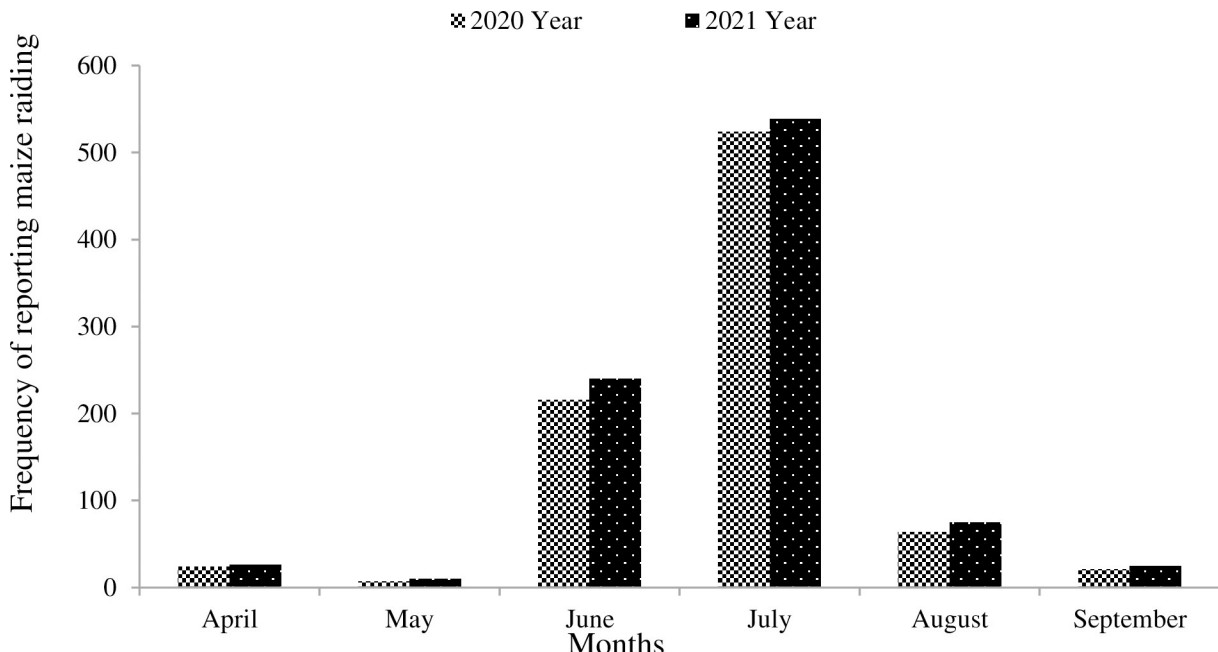

**Fig 3. The primate maize raiding frequency shows 524 ± 3.8 cobs in 2020 and 539 ± 4.6 cobs in 2021 recorded in July.** In June, 216 ± 4.6 cobs (2020) and 240 ± 5.2 cobs (2021) were recorded, while August saw 64 ± 2.1 cobs (2020) and 25 ± 1.6 cobs (2021) plucked. The lowest raiding was occurred in April and May for both years.

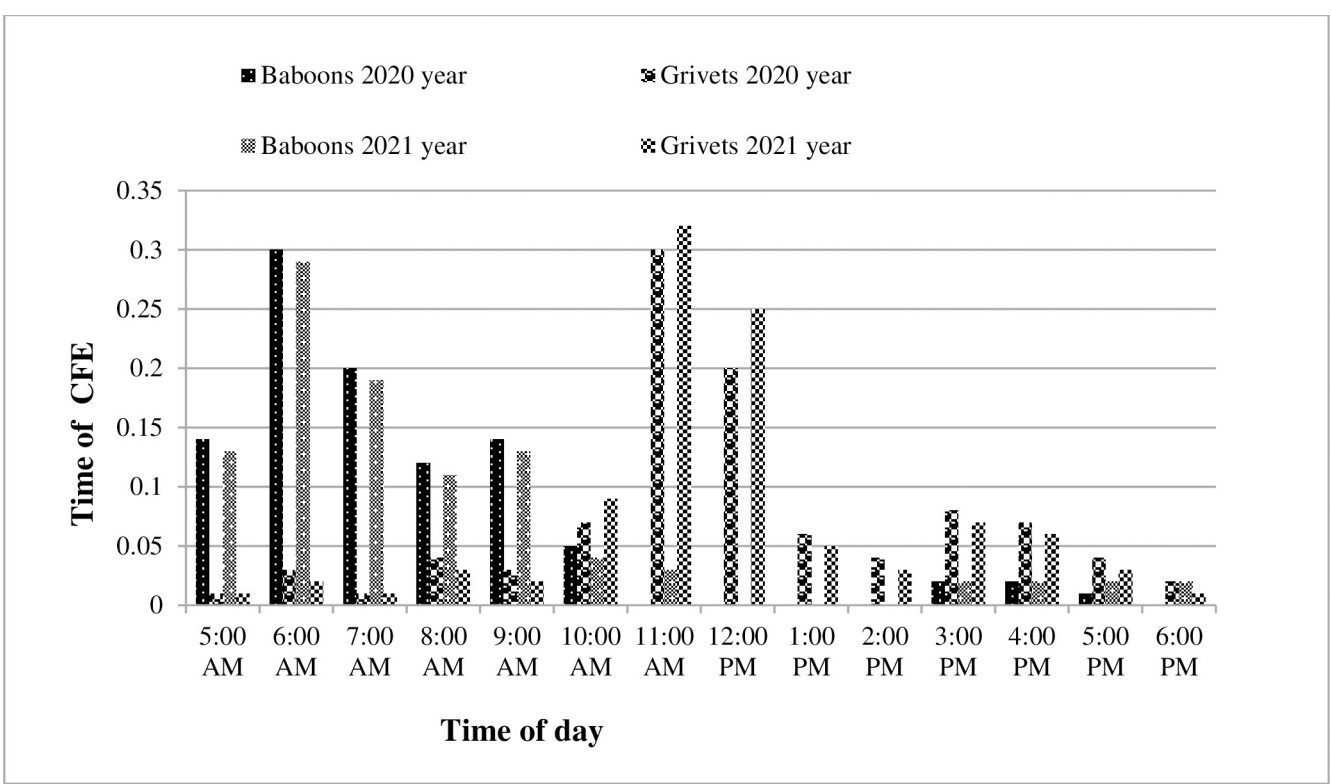

**Fig 4. The frequency of CFEs by baboons and grivet monkeys (N = 95) from April to September 2020 and 2021 shows non-uniform distributions (p < 0.001).** Baboons had more CFEs in the morning (6:00–7:00 a.m.) than in the afternoon (2:00–3:30 p.m.), while grivet monkeys peaked in the early afternoon (11:00 a.m.–12:00 p.m.). No baboon CFEs were recorded between 11:00 a.m. and 6:00 p.m. in both years.

comparison, grivet monkey raiding groups were even smaller, with 84% consisting of fewer than five individuals (S6 Fig).

A significantly greater proportion of raids (64%; n = 61) occurred in groups rather than as single raids ($\chi^2$ = 15.9, df = 4, p = 0.003). Among the group raids, 67% consisted of either 2-CRE or 3-CRE groupings, indicating a diverse pattern of multiple-CRE profiles for both grivet monkeys and baboons (Fig 5). On the other hand, single raids accounted for 36% (n = 34) and were more likely to involve a single raiding individual. It's worth noting that the extent of maize crop damage per CRE differed significantly between single raids and group raids, as evidenced by the F-test (F = 22.17, df = 1, p < 0.001). Seventy-five percent of primate field visits (comprising 22.3% olive baboons and 26.2% grivet monkeys) did not involve crop raiding (S7 Fig). Among the field visits that did include crop raiding, it was observed that 76% of olive

**Table 4. During CREs (n = 367), 75% of on-field primates ventured into the fields: 79% (n = 75 CREs) were olive baboons and 21% (n = 20 CREs) were grivet monkeys.** Olive baboons were located closer to the forest edge than grivet monkeys (p < 0.001).

| Species | Total number of individuals on fields | | | |
| --- | --- | --- | --- | --- |
| | Adults | Sub-adults | Infants | Total |
| Olive baboon | 151 (57.6%) | 78 (29.8%) | 33 (12.6%) | 262 |
| Grivet monkey | 65 (61.9%) | 40 (38.1%) | 0 (0%) | 105 |
| Total | 216 | 118 | 33 | 367 |

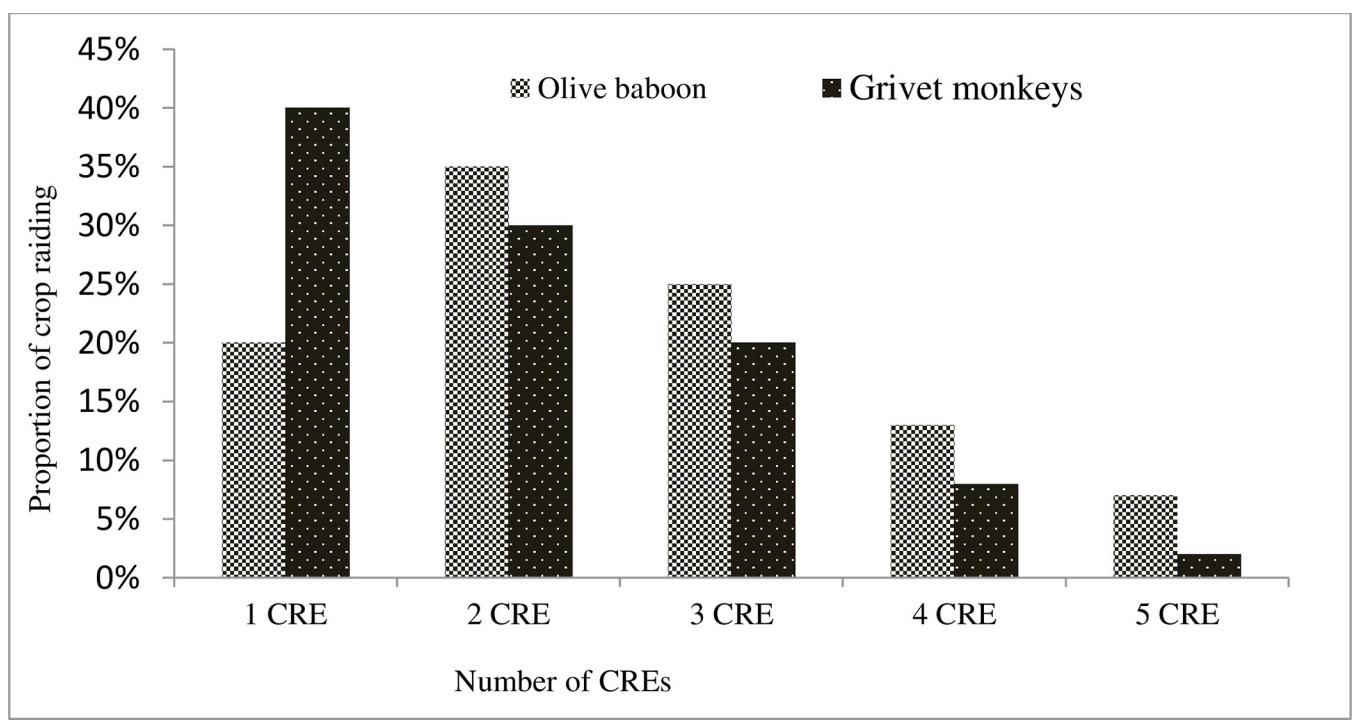

**Fig 5. The frequency distribution of CREs among primates (n = 95) shows 64% of raids occurred in groups and 36% were single raids (p = 0.003).** Among group raids, 67% involved 2-CRE or 3-CRE groupings. The extent of maize crop damage per CRE significantly differed between single and group raids (p < 0.001).

baboon visits involved multiple CREs. In the case of grivet monkeys, 53% of visits involved multiple CREs (S8 Fig).

Significantly more adults than sub-adults and more sub-adults than infants were observed in the study maize fields during CREs. These differences were statistically significant (Mann-Whitney U tests: n (sub-adult) = 118, n (adult) = 216, U = 1653.5, p < 0.001; n (infant) = 33, n (sub-adult) = 118, U = 952.0, p = 0.510). This age category distribution was consistent for each primate species ($\chi^2$ = 71.4, df = 1, p < 0.001) (Table 5). Nearly 58% (n = 55) of raiders were single adults, and the majority of adults were present in 42% of CREs involving multiple individuals (n = 40). Baboons exhibited mixed age-category raiding groups significantly more frequently than grivet monkeys (Kruskal-Wallis test, $\chi^2$ = 58.05, df = 1, p < 0.001). Most baboon and grivet raiders were accompanied by an adult during their raids. Almost two-thirds of baboon raiding groups included one or more sub-adults. Infants occasionally interacted with crops by pulling or biting stems; they often traveled or rested near an adult female. There was no significant difference between the number of male (n = 38) and female (n = 14) adult baboons observed in the fields during CREs ($\chi^2$ = 29.45, df = 1, p < 0.001). While significantly

**Table 5. During CREs (n = 95), 36% of olive baboons were adult raiders and 64% were mixed raiders.** For grivet monkeys, 68% were adult raiders and 32% were mixed raiders. This age category distribution was consistent across both species (p < 0.001).

| Species | Composition of crop-raiding group | | | |
|---|---|---|---|---|
| | Adults only | Adults and sub-adults | Adults and infants | Adults, sub-adults, infants |
| | % CREs | % CREs | % CREs | % CREs |
| Olive baboon | 36 | 45 | 4.4 | 14.6 |
| Grivet monkey | 68 | 32 | 0.0 | 0.0 |

more maize stems were damaged by mixed-age groups than by adults-only groups, the former groups also comprised more individuals, traveled further onto fields, and raided for longer durations (Mann-Whitney U tests (n (adults) = 10.0, n (mixed) = 36: stems U = 2840.5, p = 0.021; individuals U = 20.5, p = 0.367; maximum distance U = 24.5, p = 1.000; median distance U = 429.0, p = 1.000; duration U = 528.5, p < 0.001).

## Discussions

Numerous primate species have been involved in crop-raiding activities, as documented in various studies [37, 58–62]. In this study, the average maize yield loss due to primate damage was estimated at 67.9 kg per *timad* (quarter hectare), representing 43.1% in protected fields at 50 m from the forest edge. Unprotected fields experienced yield losses of 14.4%, 13.2%, 3.7%, and 0.1% at distances of 50, 100, 200, and 300 m from the forest edge, respectively. In comparison, a study by [3] reported maize yield losses of 243 kg (34.2%) and 80 kg (11.5%) per hectare due to crop-raiding by baboons and pigs in villages closer to and farther from forests, respectively. In Uganda's Budongo Forest Reserve, farmers reported that 73% of crop damage was caused by primates [9]. Additionally, in Kenya's Taita Hills, a forest-agricultural mosaic landscape, 87% of maize crops were damaged by primates [44]. The resulting average monetary losses for farmer households amounted to 1,103 ETB (equivalent to 32 US dollars), from an expected income of 8,125 ETB (equivalent to 233 US dollars) per *timad* [63].

In this study, the linear mixed model provides parameter estimates of maize crop loss during primate crop foraging events, while the fitted linear model serves as a reliable predictor for estimating the total number of crop loss events caused by wildlife [64]. Conversely, multiple regression models offer an improved estimate of maize crop loss during primate CREs by focusing on crop prevalence, with maize being most frequently raided by olive baboons and vervet monkeys [37]. Similarly, the maize model maintains broad applicability while capturing a significant proportion of local stem damage [37]. Considering that primate raiding behavior is often context-dependent [9], it is unlikely that CRE parameters contribute equally to maize crop loss during a raid [37]. This study demonstrates the value of strategically positioned camera traps in providing insights into various aspects, including recording primate species, their targeted crop types and growth phases, daily and seasonal patterns of crop-feeding activity, and whether crop-feeding occurs individually or in groups [39]. Our identifications were likely biased toward more conspicuous individuals, primarily adult males [39]. Additionally, while camera traps may capture evidence of primate groups' presence in fields, they may not consistently provide photographic evidence of actual crop manipulation and consumption [39]. Therefore, many events identified as crop feeding events through camera traps may not indeed be actual CFEs. Baboons raided the crops that are available close to the forest edge. Primates predominantly raided crops within 10 meters of the farm-forest edges [60, 65, 66]. However, baboons still visited farms located 300 meters from the forest edge, even though maize crop feeding events were infrequent at this distance. In Uganda, vervet monkeys ventured up to 55 meters into crop fields, while baboons reached up to 110 meters [67]. The highest distance observed was over 700 meters, notably in the Ngangao Forest in the Taita Hills, Kenya [44]. This variation may be influenced by the distribution of households and the number of farms investigated at different distances [44].

In this study, maize raids by primates were observed during the maturation of maize cobs. The findings suggest that scarecrows and thorn bushes were generally ineffective in preventing baboons and grivet monkeys from returning to the fields. While wire mesh protection reduced maize damage, it did not fully deter baboons, as they quickly habituated to it. At the Kokate Marachare site, the wire mesh fence was somewhat effective in discouraging olive baboons and

grivet monkeys, likely due to the presence of a single raider. However, at the Gurumu Woide site, where multiple baboons were present, they remained vigilant and determined to raid the maize crops, even though the fields were fenced with wire mesh. Similarly, wire mesh fences showed limited effectiveness against primate raiding in the Budongo Forest Reserve, Uganda [42]. Indeed, field guards were often absent due to other (social) activities, school attendance, etc. However, continuous guarding is a key strategy for effectively mitigating crop damage by pests [3]. The extended protection duration was particularly necessary in villages at higher altitudes where maize takes longer to mature [3]. Both olive baboons and grivet monkeys are frequently observed foraging for crops in human-dominated settings in the study area, with olive baboons causing more damage than grivet monkeys. Similarly, olive baboons and vervet monkeys in the study area were damaging maize crops through feeding, trampling, and the destruction of stems and roots. This has significantly impacted maize yields and household incomes. Despite the abundance of forest fruits, the primates' appetite for maize remains undiminished, and they continue to forage on the crops [68].

The time of day had differing effects on the crop-foraging patterns of the two species, with olive baboons foraging more frequently in the morning and grivet monkeys in the afternoon. This variation in the time of activity might be related to the presence of baboons, which appeared to deter grivet crop-foraging behavior [69]. Similarly, the time activity pattern varied in different areas; [70] recorded a peak in baboon crop foraging in Zimbabwe between 8 and 10 am, potentially driven by the need to find food upon walking. In contrast, primates in Uganda foraged on crops more frequently between noon and sunset than between sunrise and noon [9].

To access crops, baboons were observed using a 'sit and wait' strategy near the edge of crop fields [71]. The more time olive baboons and grivet monkeys spent close to the fields, the more probability they were to forage crops. Furthermore, when they entered crops during these visits, they were more likely to enter multiple times. Crop raiding was not a foraging pattern practiced by all members of primate social groups, with baboon raiding parties typically averaging five individuals [60].

In this study, more adults were observed on maize fields during CREs compared to sub-adults. This varies in different areas; in some studies, adult primates were the main crop raiders, as referenced in [60–62, 69], while in other studies, sub-adults were identified as the primary raiders, as cited in [72–75]. However, this behavior was rare and observed only in baboons [37]. Additionally, perceptions of risk may influence the age composition of primate raiding groups, with adult females accompanied by infants raiding less frequently, likely due to increased caution [62, 76].

## Conclusion

The significant crop losses observed underscore the need for continuous vigilance in maize fields, from sowing to harvest, to deter wild primate pests. The parameters of crop foraging events can serve as quantifiable measures for assessing the effectiveness of various techniques aimed at deterring primate crop foraging. In this study, wire mesh fencing and guarding were found to have limited effectiveness in preventing raids by olive baboons and grivet monkeys. Therefore, no single mitigation method proved completely effective in preventing primate crop raiding during this study, implying the need to apply a combination of mitigation strategies. The participatory approach, combined with camera traps, was proven to be an appropriate method for assessing primate-induced maize damage. The linear mixed model (LMM) was a suitable choice for analyzing the extent of maize damage by primates across various spatio-temporal factors. Understanding the spatio-temporal patterns of wildlife-induced crop losses,

as well as evaluating key parameters related to crop foraging events, is essential for mitigating the socio-economic impacts of primate pests originating from forest edges.

## Supporting information

**S1 Fig. Location map of the study area (created with ESRI ArcGIS Desktop 10.8).**
(TIF)

**S2 Fig.** Various prevention strategies (wire mesh (A), human guardian tower (B), scarecrow (C), and thorny bush (D)) were assessed in eight experimental maize field sites to evaluate their effectiveness in deterring crop raiders. The study was conducted in maize field sites located in Gurumu Woide and Kokate Marachare (Photo credit: Yigrem Deneke).
(TIF)

**S3 Fig. Diagrammatic example of a field map used by observers.** HSE = house. GH = guard hut. SH = storage hut. Solid black lines = field boundary. Green objects = trees.
(TIF)

**S4 Fig.** The images above depict camera trap captures of various wildlife species observed in maize field sites located in Damota Mountain, Southern Ethiopia: (A) Olive baboons (*Papio anubis*); (B) Grivet monkeys (*Chlorocebus aethiops*); (C) Porcupines (*Hystrix cristata*); and (D) Bushbucks (*Tragelaphus scriptus*).
(TIF)

**S5 Fig. Relative frequency of raid durations by primate CREs (n = 95).**
(TIF)

**S6 Fig. Relative frequency of raiding by primate CREs (n = 95).**
(TIF)

**S7 Fig. The number of baboon and grivet monkey field visits that did and did not involve crop-raiding events on maize fields in the highlands of Damota mountain, April to September 2020 and 2021 years (n = 367).**
(TIF)

**S8 Fig. The number of baboon and grivet monkey field visits that involved single- and multi-crop raiding events on maize fields in the highlands of Damota mountain, April to September 2020 and 2021 years (n = 189).**
(TIF)

**S1 Table. Maize field and study plot size on the protective and non-protective maize fields.**
(DOCX)

**S2 Table. Farmer observation and reported of maize damage assessments (580 maize stem expected per plot except field no. 25 (see the text).**
(DOCX)

**S3 Table. Camera traps recorded the CREs and CFEs of olive baboons and grivet monkeys among twenty-five selected maize fields.** Each field comprised study plots measuring 10mx10 meters, observed during the maize cropping seasons of 2020 and 2021.
(DOCX)

**S4 Table. A linear mixed model of the maize damage rate by primates, considering different spatio-temporal variables, was analyzed using R code.**
(DOCX)

**S1 File. The rate of maize damage by olive baboons in different crop phonological stages was analyzed in both protected and open/control fields using R code.**
(DOCX)

**S2 File. The rate of maize damage by grivet monkeys in different crop phenological stages was analyzed in both protected and open/control fields using R code.**
(DOCX)

**S3 File. English language proficiency 1st editor certificate.**
(PDF)

**S4 File. English language proficiency 2nd editor certificate.**
(PDF)

**S1 Video. Maize crop foraging events by olive baboons, October 15, 2021 at 7:42am (UTC).**
(AVI)

## Acknowledgments

We would like to express our gratitude to VLIR-UOS, the University of Antwerp in Belgium, and Wolaita Sodo University in Ethiopia for their technical and administrative support. We are also grateful to the Environment Protection, Forest and Climate Change Regulatory Office of Wolaita Zone, Ethiopia, for granting us permission to conduct this research. Additionally, we extend our thanks to the administrators of the respective districts and villages, as well as the agriculture and rural development office workers of Sodo Zuriya and Damot Gale, and the local farmers for their unwavering technical support, enthusiasm, and hospitality throughout our research. Finally, we would like to thank all the reviewers and editors for their valuable comments, which greatly improved this manuscript.

## Author Contributions

**Conceptualization:** Yigrem Deneke.

**Data curation:** Yigrem Deneke.

**Formal analysis:** Yigrem Deneke.

**Investigation:** Yigrem Deneke, Aberham Megaze, Wondimagegnehu Tekalign, Taye Dobamo.

**Methodology:** Yigrem Deneke.

**Project administration:** Herwig Leirs.

**Resources:** Herwig Leirs.

**Supervision:** Aberham Megaze, Herwig Leirs.

**Writing – original draft:** Yigrem Deneke.

**Writing – review & editing:** Yigrem Deneke, Aberham Megaze, Herwig Leirs.

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
