## [Decision Letter · Decision Letter 0]

9 Feb 2024

PONE-D-23-43746Crop Damage by Nonhuman Primates: Quantifying the Keys Parameters of Crop-Raiding events on the Livelihoods of Smallholders in an Agriculture- Forest Mosaic Landscape, Wolaita Zone, Southern EthiopiaPLOS ONE

Dear Dr. Deneke,

Thank you for submitting your manuscript to PLOS ONE. After careful consideration, we feel that it has merit but does not fully meet PLOS ONE’s publication criteria as it currently stands. Therefore, we invite you to submit a revised version of the manuscript that addresses the points raised during the review process.

We look forward to receiving your revised manuscript.

Kind regards,

Sharon E Kessler

Academic Editor

PLOS ONE

Journal Requirements:

3. Please be informed that funding information should not appear in the Acknowledgments section or other areas of your manuscript. We will only publish funding information present in the Funding Statement section of the online submission form. Please remove any funding-related text from the manuscript. 

6. Please ensure that you refer to Figure 2 in your text as, if accepted, production will need this reference to link the reader to the figure.

7. We note that Figure 1 in your submission contain map images which may be copyrighted. All PLOS content is published under the Creative Commons Attribution License (CC BY 4.0), which means that the manuscript, images, and Supporting Information files will be freely available online, and any third party is permitted to access, download, copy, distribute, and use these materials in any way, even commercially, with proper attribution. For these reasons, we cannot publish previously copyrighted maps or satellite images created using proprietary data, such as Google software (Google Maps, Street View, and Earth). For more information, see our copyright guidelines: http://journals.plos.org/plosone/s/licenses-and-copyright.

(1) You may seek permission from the original copyright holder of Figure 1 to publish the content specifically under the CC BY 4.0 license.  

**Additional Editor Comments:**

I have now received two reviews of your paper. Reviewer 1 has pointed out some substantial revisions that need to be made before the paper can be considered for publication. Therefore, I would invite you to engage in major revisions. Please address all of the reviewers comments. In particular, please provide a more substantial grounding for your study in the context of the literature in your introduction and discussion. Please also consider the more comprehensive approaches to your data that reviewer 1 has suggested. I look forward to reading your revised manuscript.

Reviewers' comments:

Reviewer's Responses to Questions

**Comments to the Author**

1. Is the manuscript technically sound, and do the data support the conclusions?

Reviewer #1: Partly

Reviewer #2: Yes

2. Has the statistical analysis been performed appropriately and rigorously? 

Reviewer #1: No

Reviewer #2: Yes

3. Have the authors made all data underlying the findings in their manuscript fully available?

Reviewer #1: No

Reviewer #2: Yes

4. Is the manuscript presented in an intelligible fashion and written in standard English?

Reviewer #1: No

Reviewer #2: Yes

5. Review Comments to the Author

Reviewer #1: Title: Please make your title concise and precise; it is now unnecessarily long and complex

I would suggest to be modified as: ‘Patterns of Primates’ Crop Raiding and the impacts on incomes of Smallholders across Mosaic agricultural Landscape of Wolaita Zone, Southern Ethiopia’

Introduction: this section is too shallow, lacks strong evidence from bodies of literature. Instead, it seems the description of the study area. Hence, I would recommend major revisions by scanning more literature review. For example, you can refer: Lemessa, D., Hylander, K., & Hambäck, P. (2013). Composition of crops and land-use types in relation to crop raiding pattern at different distances from forests. Agriculture, Ecosystems and Environment, 167, 71–78. https://doi.org/10.1016/j.agee.2012.12.014

Materials and methods

Study area

Please include the ecosystem type and farming systems in the description of the study areas. Refer to this doc regarding the farming system: Amede, T., Auricht, C., Boffa, J.-M., Dixon, J., Mallawaarachchi, T., Rukuni, M., & Teklewold-Deneke, T. (2015). The Evolving Farming and Pastoral Landscapes in Ethiopia: A Farming System Framework for Investment Planning and Priority Setting. September.

Line 146-148, how can famers identify the age categories of the crop raiders? It is not clear!

Experimental setup: you need to show experimental setup by drawing schematic sketch, it is not entirely clear from the texts you wrote??

Data analysis

This is entirely less clear and has a lot of shortfalls. The study was undertaken across a landscape and hence, it is expected that there random factors besides the data considered as fixed factors. The data analysis undertaken did not take into account this issue. This means that such dataset can strongly be analyzed using mixed models such as LMM or GLMM. Now, simple non-parametric statistical tests were used and even are not clearly described how and for what type of response variables they used. For example, Ch-square for what kind of frequency distribution? Mann Whitney test for what type of data? and F-test?

Line 188-190, For the analysis of primate assaults on maize within both preventive and non-preventive maize fields at different seasons and crop phenology, we utilized R-Software [13]. But, what package and function were used is not mentioned.

Moreover, there are statistical tools mentioned in the result but not in data analysis section. Why the authors used two statistical software is also not clear? Hence, I would strongly suggest to run the analysis based on these comments and reciprocally present the results.

Results

I was a bit confused how the authors described the crop damage from maize stems and cobs.

- Is it stems/cob or cobs/stems?

Discussion section

Since some results may be changed after the revision of the analysis, at this stage it is not sensible to comment on this section.

Reviewer #2: Comments and suggestions to the Authors

This paper is good and can be strengthened for your audience to benefit even more from your study. I made some minor comments and suggestions. The authors should use the separate file to correct the suggested comments. In addition, there are some comments and suggestions that require clarification.

Title:

1. The title is very long. It should be shorter than the present and to the point accordingly.

2. The title is similar with the work titled “Crop Damage by Primates: Quantifying the Key Parameters of Crop-Raiding Events” in western Uganda and published in PLOS ONE although it is in different countries. Would you modify your title?

Abstract:

1. Line -13- “This study aims to assess primates foraging behavior” Is assessing primates foraging behavior was one of the objective of the paper? If yes where is the method and result of the objective?

2. An abstract should show slightly how to collect the data of the work. However, this manuscript didn’t show how to collect the data.

Introduction

1. The authors described as the study area contains endemic animal and plant species. Would you please list some of them?

2. The gap of the study should describe at the introduction part. However, I didn’t see a research gap in your introduction section.

3. Line-39- 47- It will be better if this part moved to the description of the study area section.

4. Lines -55- Please remove the method from the introduction part.

5. The introduction part is very short. It didn’t highlight what is done and what is the gap. I recommend writing strong introduction.

Study area

1. Line-68- remove the word “see”

Experimental setup

1. What do 25 fields mean? Would you clarify in detail?

2. Line -122- “Data collection was carried out by farmers who had received training from researchers”. I doubt about this. How farmers can collect data? The work is on crop damage by Nonhuman Primates. If yes how farmers can collect actual data. I strongly believe that the data collected by farmers might be biased. Why didn’t participate field experts rather than the farmers.

3. Line-127-128- The data collectors (farmers) received incentives. However, this may makes them biased on the data.

4. I have doubt on the data collection method (Farmer observation and reports). Thus, the authors should give scientific evidence on the doubt why they use farmers as data collectors by giving incentives.

Data analysis

Result

1. Line-213- remove the word “as detailed in”

2. The result of the manuscript is very shallow. Please try to make it strong result.

Discussion

1. Line -349- 354- this is a repetition of the result. Please don’t repeat your result in the discussion section. Instead try to discuss your result (the meaning and application of your result).

2. Your discussion is very broad and general. I suggest you to give emphasis on the interpretation of your result and comparing your result with recent publications similar to your work.

Conclusion

1. Would you please add the application of your work in the introduction section?

2. Make it strong conclusion based on your findings.

General Comments

1. Grammar and structure of sentences should be revised for better improvement.

I hope this helps to make the paper better.

Best wishes.

6. PLOS authors have the option to publish the peer review history of their article (what does this mean?). If published, this will include your full peer review and any attached files.

Reviewer #1: **Yes: **Debissa Lemessa

Reviewer #2: No

---

## [Author Response · Author response to Decision Letter 0]

3 Mar 2024

Title: Crop Damage by Nonhuman Primates: Quantifying the Keys Parameters of Crop Raiding events on the Livelihoods of Smallholders in an Agriculture- Forest Mosaic Landscape, Wolaita Zone, Southern Ethiopia

Manuscript Number: PONE-D-23-43746

Response to Reviewers

We are great full to the editors and reviewers for their insightful and valuable comments on our paper. We have carefully considered the comments and tried our best to address every one of them. We hope the manuscript after careful revisions meet your higher standard journal. The authors welcome further constructive comments if any. We provided the point by point responses. All modifications in the manuscript have been highlighted in yellow color.

Sincerely,

Yigrem Deneke

Response to Reviewer 1

Title: Please make your title concise and precise; it is now unnecessarily long and complex

I would suggest to be modified as: ‘Patterns of Primates’ Crop Raiding and the impacts on incomes of Smallholders across Mosaic agricultural Landscape of Wolaita Zone, Southern Ethiopia’

Response: The title has been modified to "Patterns of Primates Crop Raiding and the Impacts on Incomes of Smallholders across Mosaic Agricultural Landscape of Wolaita Zone, Southern Ethiopia."

Introduction: this section is too shallow, lacks strong evidence from bodies of literature. Instead, it seems the description of the study area. Hence, I would recommend major revisions by scanning more literature review. For example, you can refer: Lemessa, D., Hylander, K., & Hambäck, P. (2013). Composition of crops and land-use types in relation to crop raiding pattern at different distances from forests. Agriculture, Ecosystems and Environment, 167, 71–78. https://doi.org/10.1016/j.agee.2012.12.014

Response: The introduction has been revised according to the comments and some points are move to description of the study area.

Materials and methods

Study area

Please include the ecosystem type and farming systems in the description of the study areas. Refer to this doc regarding the farming system: Amede, T., Auricht, C., Boffa, J.-M., Dixon, J., Mallawaarachchi, T., Rukuni, M., & Teklewold-Deneke, T. (2015). The Evolving Farming and Pastoral Landscapes in Ethiopia: A Farming System Framework for Investment Planning and Priority Setting. September.

Response: The introduction has been revised to incorporate the comments:

"The Wolaita zone, characterized by a highland perennial farming system, supports a diverse array of crops (Amede et al., 2017). Primary food crops in this region, as reported by Amede et al. (2017), include maize, teff, various vegetables, and root and tuber species such as cassava, yam, potato, sweet potato, and taro. Additionally, tropical and temperate fruit tree crops like banana, avocado, mango, and apple are cultivated in the Wolaita areas (Amede et al., 2017)."

Line 146-148, how can famers identify the age categories of the crop raiders? It is not clear!

Response: Most of the farmers had completed secondary school, and some of them had received bachelor's degrees in related fields such as plant science and biology. They possess indigenous knowledge regarding the identification of primate species and age categories of the crop raiders. Additionally, they were trained by the researchers to become more familiar with using scientific methods for identifying the age categories of the crop raiders.

Experimental setup: you need to show experimental setup by drawing schematic sketch, it is not entirely clear from the texts you wrote??

Response: I depicted the experimental setup by including a schematic sketch in this manuscript.

Data analysis

This is entirely less clear and has a lot of shortfalls. The study was undertaken across a landscape and hence, it is expected that there random factors besides the data considered as fixed factors. The data analysis undertaken did not take into account this issue. This means that such dataset can strongly be analyzed using mixed models such as LMM or GLMM. Now, simple non-parametric statistical tests were used and even are not clearly described how and for what type of response variables they used. For example, Ch-square for what kind of frequency distribution? Mann Whitney test for what type of data? and F-test?

Response: We chose a multiple regression model over LMM or GLMM models because it is suitable for analyzing spatio-temporal data of the dependent variable (maize rate of damage) along with multiple independent variables, such as the number of individuals raiding, primate CREs, field distance, duration of raiding, and crop phenology. This statistical model is supported by Wallace and Hill (2012). A chi-square test was employed to examine the variation in the amount of maize damage by primates across different variables, including primate species raiding duration, multiple versus single raid events, primate CFE in time of day, and age-category of raiding in single or group. The Mann-Whitney U test was utilized to compare primate CREs of raiding durations and different age categories of primate species on CREs. Additionally, the F-test was applied to compare estimates of maize damage between preventive and non-preventive strategies, as well as between single and multiple raids.

Line 188-190, For the analysis of primate assaults on maize within both preventive and non-preventive maize fields at different seasons and crop phenology, we utilized R-Software [13]. But, what package and function were used is not mentioned. Moreover, there are statistical tools mentioned in the result but not in data analysis section. Why the authors used two statistical software is also not clear? Hence, I would strongly suggest to run the analysis based on these comments and reciprocally present the results.

Response: In R software, we utilized the bplot function in the Rlab package. We found that creating graphics such as boxplots is more effective and easier to use for data interpretation compared to SPSS. Therefore, we employed two statistical tests.

Results

I was a bit confused how the authors described the crop damage from maize stems and cobs.

- Is it stems/cob or cobs/stems?

Response: It was observed that maize stems were damaged and cobs were plucked (stems/cobs).

Discussion section

Since some results may be changed after the revision of the analysis, at this stage it is not sensible to comment on this section. 

Response: Ok. It is modified based on results

Response to Reviewer 2

This paper is good and can be strengthened for your audience to benefit even more from your study.

Response: Thank you very much.

Title

1. The title is very long. It should be shorter than the present and to the point accordingly.

2. The title is similar with the work titled “Crop Damage by Primates: Quantifying the Key Parameters of Crop-Raiding Events” in western Uganda and published in PLOS ONE although it is in different countries. Would you modify your title?

Response: It is modified 

Abstract:

1. Line -13- “This study aims to assess primates foraging behavior” Is assessing primates foraging behavior was one of the objective of the paper? If yes where is the method and result of the objective? 

2. An abstract should show slightly how to collect the data of the work. However, this manuscript didn’t show how to collect the data.

Response: I have modified it to "primate crop foraging/raiding events (CFE/CRE)." The method is described in the manuscript, and the data collection was conducted through farmer observation and reports, as well as camera traps. The data collection method is also incorporated into the abstract.

Introduction

1. The authors described as the study area contains endemic animal and plant species. Would you please list some of them?

2. The gap of the study should describe at the introduction part. However, I didn’t see a research gap in your introduction section. 

3. Line-39- 47- It will be better if this part moved to the description of the study area section.

4. Lines -55- Please remove the method from the introduction part. 

5. The introduction part is very short. It didn’t highlight what is done and what is the gap. I recommend writing strong introduction.

Response: The introduction has been revised based on the comments. The research gap is indicated, and the description of endemic animal and plant species, as well as the detailed description of the study area, has been removed from the manuscript.

Study area

1. Line-68- remove the word “see”

Response: I removed the word “see” 

Experimental setup

1. What do 25 fields mean? Would you clarify in detail?

2. Line -122- “Data collection was carried out by farmers who had received training from researchers”. I doubt about this. How farmers can collect data? The work is on crop damage by Nonhuman Primates. If yes how farmers can collect actual data. I strongly believe that the data collected by farmers might be biased. Why didn’t participate field experts rather than the farmers.

3. Line-127-128- The data collectors (farmers) received incentives. However, this may makes them biased on the data.

4. I have doubt on the data collection method (Farmer observation and reports). Thus, the authors should give scientific evidence on the doubt why they use farmers as data collectors by giving incentives. 

Response: Twenty-five maize fields were selected for this study, comprising 8 protective and 17 non-protective maize fields. Most of the farmers involved had completed secondary school, and some of them had received bachelor's degrees in related fields. The farmers managed their own maize fields during the day and night, drawing on their indigenous knowledge and practices concerning human-wildlife conflict assessments. All these farmer participants in data collection are members of Damota Community Managed Forest, and they are developing knowledge and skills from the forest managers and local NGOs (World Vision of Ethiopia) on how to manage and protect the forest, as well as assessing human-wildlife conflict and mitigating strategies. Additionally, all farmers were trained by the researchers to familiarize themselves more with scientific data collection protocol. Moreover, researchers supervised the data collection process. The farmers received per diem payments for guarding the camera traps during four consecutive trapping nights per month, and they also received compensation fees at the end of the project period.

Data analysis 

Result

1. Line-213- remove the word “as detailed in”

2. The result of the manuscript is very shallow. Please try to make it strong result.

Response: I removed the phrase "as detailed in." Additionally, I included an additional result that highlights the strength of using multiple regression models for the analysis.

Discussion

1. Line -349- 354- this is a repetition of the result. Please don’t repeat your result in the discussion section. Instead try to discuss your result (the meaning and application of your result). 

2. Your discussion is very broad and general. I suggest you to give emphasis on the interpretation of your result and comparing your result with recent publications similar to your work. 

Response: I have revised this section by eliminating the repetition of the result. Additionally, I compared our findings with those of similar works, including recent publications.

Conclusion

1. Would you please add the application of your work in the introduction section? 

2. Make it strong conclusion based on your findings.

Response: I have described this section based on my findings.

Grammar and structure of sentences should be revised for better improvement.

Response: Grammar and structure of sentences have been improved

---

## [Decision Letter · Decision Letter 1]

1 Apr 2024

PONE-D-23-43746R1Crop Damage by Nonhuman Primates: Quantifying the Keys Parameters of Crop-Raiding events on the Livelihoods of Smallholders in an Agriculture- Forest Mosaic Landscape, Wolaita Zone, Southern EthiopiaPLOS ONE

Dear Dr. Deneke,

Thank you for submitting your manuscript to PLOS ONE. After careful consideration, we feel that it has merit but does not fully meet PLOS ONE’s publication criteria as it currently stands. Therefore, we invite you to submit a revised version of the manuscript that addresses the points raised during the review process. Both reviewers have re-reviewed your manuscript and have indicated that they are not satisfied with the revisions. Therefore, I invite you to revise again and to address all of their concerns. In particular, please include a more thorough literature review in your introduction, please address the statistical concerns of reviewer 1 (I would encourage you to use the recommended analysis), and please discuss the methodological concerns of reviewer 2 (points 2 & 3, which the reviewer refers to as 'serious.')  I look forward to reading the revised manuscript.

We look forward to receiving your revised manuscript.

Kind regards,

Sharon E Kessler

Academic Editor

PLOS ONE

Reviewers' comments:

Reviewer's Responses to Questions

**Comments to the Author**

1. If the authors have adequately addressed your comments raised in a previous round of review and you feel that this manuscript is now acceptable for publication, you may indicate that here to bypass the “Comments to the Author” section, enter your conflict of interest statement in the “Confidential to Editor” section, and submit your "Accept" recommendation.

Reviewer #1: (No Response)

Reviewer #2: (No Response)

2. Is the manuscript technically sound, and do the data support the conclusions?

Reviewer #1: Yes

Reviewer #2: Yes

3. Has the statistical analysis been performed appropriately and rigorously? 

Reviewer #1: No

Reviewer #2: Yes

4. Have the authors made all data underlying the findings in their manuscript fully available?

Reviewer #1: Yes

Reviewer #2: Yes

5. Is the manuscript presented in an intelligible fashion and written in standard English?

Reviewer #1: Yes

Reviewer #2: No

6. Review Comments to the Author

Reviewer #1: 1. ‘it is easier to make visualizations of the results, e.g. plotting box plot using SPSS instead of R’, but you have used R to run lm analysis which is even more difficult when compared to making box plot graph using e.g., boxplot<-boxplot(damage~ locations). This does not convince me and I strongly suggest using R instead of SPSS to produce high quality graph.

2. ‘We chose a multiple regression model over LMM or GLMM models because it is suitable for analyzing spatio-temporal data of the dependent variable (maize rate of damage)’ over the LMM. However, you have random factors which, due to the variations among sites/seasons, unnecessarily inflate the impacts of noises in the model and subsequently the results. To intuitively analyze the data and thereby enhance the novelty of your study, I still insist to run the analysis using LMM which is the extension of lm (that you have used now) to take into account both fixed factors (that you measured or recorded) and the random factors (sites/seasons) that you cannot do this with ordinary linear model.

Reviewer #2: Comments and suggestions to the Authors

Although most of the previous comments and suggestions are corrected, there are still some comments and queries that require improvements and clarification.

Title

1. Modified according the comments

Abstract:

1. I didn’t see the highlight how to collect the data of the work in the abstract section. Yes, it is described in detail in the MS. However, abstract should highlight the method of data collection.

Introduction

1. Corrected and improved.

Study area

1. Line 107, According to reference [25], delete the word reference

2. Line 13 and 111, 10x10 meters, correct as 10 x10 meters

Farmer observation and reports

1. Farmer? Only one farmer or farmers?

2. Line -141- “The data collection was conducted by twenty five farmers who had received training from researchers…..” Still, this concern is serious. How farmers can collect data? Although they completed secondary school or above, they are already farmers. Thus, how farmers can collect actual data. This may lead to biasness. Why didn’t participate field experts rather than the farmers?

3. Furthermore, the data the farmers received incentives. However, this may makes them biased on the data.

4. Dear authors, you didn’t give a scientific response on the above two serious issues (2 and 3).

Results

1. In the results in the testing of significance level, there is writing error. For instance, line 288: (F=292.5, df=11, p < .001, see Figure 5). Here, it should rewrite as follows: (F = 292.5, df = 11, p < .001, see Figure 5). Your writing style should consistency throughout the MS.

2. There is inconsistency in the writing style of the MS. For example, Line 288 (F=292.5, df=11, p < .001, see Figure 5) and line 299 (χ² = 58.62, d.f. = 10, P < 0.05). Here, look the degree of freedom (df). In the first degree of freedom written as df while in the second degree of freedom written as d.f. You have to correct such inconsistencies throughout the manuscript.

General Comments

1. There is inconsistency in the writing style in the MS (as described as the results section in the above).

2. Furthermore, I have observed other inconsistencies in the MS. For instance, line 77 (Fig. 1), line 118 (Figure 2), line 124 (see figure 3), line 280 (Fig. 5), line 285 (as illustrated in Fig. 6), line 295 (as illustrated in Fig 7), line 303 (see Fig 8) …… Please correct these and other inconsistencies throughout the manuscript.

Accordingly, the MS requires serious editing once again before go to the production for publication.

7. PLOS authors have the option to publish the peer review history of their article (what does this mean?). If published, this will include your full peer review and any attached files.

Reviewer #1: No

Reviewer #2: No

---

## [Author Response · Author response to Decision Letter 1]

10 Apr 2024

Title: Patterns of Primates Crop Raiding and the impacts on incomes of Smallholders across Mosaic agricultural Landscape of Wolaita Zone, Southern Ethiopia 

Manuscript Number: PONE-D-23-43746-R2

Response to Reviewers

We are great full to the editors and reviewers for their insightful and valuable comments on our paper. We have carefully considered the comments and tried our best to address every one of them. We hope the manuscript after careful revisions meet your higher standard journal. The authors welcome further constructive comments if any. We provided the point by point responses. All modifications in the manuscript have been highlighted in yellow color.

Sincerely,

Yigrem Deneke

Response to Reviewer 1

Introduction

Q1. The authors have now made revisions and enhanced the clarities of their manuscript. However, still the introduction is too shallow or very short at least.

Response: Thank you very much. The introduction has been revised based on the comments and it is included additional literature review on the subject

Data analysis

Q2. ‘it is easier to make visualizations of the results, e.g. plotting box plot using SPSS instead of R’, but you have used R to run lm analysis which is even more difficult when compared to making box plot graph using e.g., boxplot<-boxplot(damage~ locations). This does not convince me and I strongly suggest using R instead of SPSS to produce high quality graph.

Response: I have revised it and making plotting box plot using SPSS 

Q3. ‘We chose a multiple regression model over LMM or GLMM models because it is suitable for analyzing spatio-temporal data of the dependent variable (maize rate of damage)’ over the LMM. However, you have random factors which, due to the variations among sites/seasons, unnecessarily inflate the impacts of noises in the model and subsequently the results. To intuitively analyze the data and thereby enhance the novelty of your study, I still insist to run the analysis using LMM which is the extension of lm (that you have used now) to take into account both fixed factors (that you measured or recorded) and the random factors (sites/seasons) that you cannot do this with ordinary linear model.

Response: I revised it and I run the analysis using LMM by taking in to account both fixed factors (that you measured or recorded) and the random factors (sites/seasons)

If he authors willing to revise their manuscript based on these points, I think the paper can be relevant to be published in PLOS ONE journal.

Response: Thank you very much.

Response to Reviewer 2

Although most of the previous comments and suggestions are corrected, there are still some comments and queries that require improvements and clarification.

Response: Thank you very much.

Title

1. Modified according the comments

Response: Thank you very much.

Abstract:

1. I didn’t see the highlight how to collect the data of the work in the abstract section. Yes, it is described in detail in the MS. However, abstract should highlight the method of data collection. 

Response: I highlight the method of data collection as per the comment.

Introduction

1. Corrected and improved. 

Response: Thank you very much.

Study area

1. Line 107, According to reference [25], delete the word reference 

2. Line 13 and 111, 10x10 meters, correct as 10 x10 meters

Response: It is modified 

Farmer observation and reports

1. Farmer? Only one farmer or farmers? 

2. Line -141- “The data collection was conducted by twenty five farmers who had received training from researchers…..” Still, this concern is serious. How farmers can collect data? Although they completed secondary school or above, they are already farmers. Thus, how farmers can collect actual data. This may lead to biasness. Why didn’t participate field experts rather than the farmers?

3. Furthermore, the data the farmers received incentives. However, this may makes them biased on the data.

4. Dear authors, you didn’t give a scientific response on the above two serious issues (2 and 3). 

Response: It is modified the method of the data collection in the manuscript based on the given comments 

Results

1. In the results in the testing of significance level, there is writing error. For instance, line 288: (F=292.5, df=11, p < .001, see Figure 5). Here, it should rewrite as follows: (F = 292.5, df = 11, p < .001, see Figure 5). Your writing style should consistency throughout the MS.

2. There is inconsistency in the writing style of the MS. For example, Line 288 (F=292.5, df=11, p < .001, see Figure 5) and line 299 (χ² = 58.62, d.f. = 10, P < 0.05). Here, look the degree of freedom (df). In the first degree of freedom written as df while in the second degree of freedom written as d.f. You have to correct such inconsistencies throughout the manuscript. 

Response: It is modified. I consistently and properly used the testing of significance level or degree of freedom (df) throughout the manuscript.

 English grammer is also improved throughout the manuscript

General Comments

1. There is inconsistency in the writing style in the MS (as described as the results section in the above).

2. Furthermore, I have observed other inconsistencies in the MS. For instance, line 77 (Fig. 1), line 118 (Figure 2), line 124 (see figure 3), line 280 (Fig. 5), line 285 (as illustrated in Fig. 6), line 295 (as illustrated in Fig 7), line 303 (see Fig 8) …… Please correct these and other inconsistencies throughout the manuscript. 

Response: It is modified. I consistently and properly wrote the figure and table on the manuscript through adhered to the journal guidelines regarding figures and table formatting style.

Accordingly, the MS requires serious editing once again before go to the production for publication.

Response: Ok, thank you very much.

---

## [Decision Letter · Decision Letter 2]

31 Jul 2024

PONE-D-23-43746R2Patterns of Primates Crop Raiding and the impacts on incomes of Smallholders across Mosaic agricultural Landscape of Wolaita Zone, Southern EthiopiaPLOS ONE

Dear Dr. Deneke,

Thank you for submitting your manuscript to PLOS ONE. After careful consideration, we feel that it has merit but does not fully meet PLOS ONE’s publication criteria as it currently stands. Therefore, we invite you to submit a revised version of the manuscript that addresses the points raised during the review process.

We look forward to receiving your revised manuscript.

Kind regards,

Miquel Vall-llosera Camps

Senior Staff Editor

PLOS ONE

Journal Requirements:

Reviewers' comments:

Reviewer's Responses to Questions

**Comments to the Author**

1. If the authors have adequately addressed your comments raised in a previous round of review and you feel that this manuscript is now acceptable for publication, you may indicate that here to bypass the “Comments to the Author” section, enter your conflict of interest statement in the “Confidential to Editor” section, and submit your "Accept" recommendation.

Reviewer #1: (No Response)

Reviewer #2: All comments have been addressed

2. Is the manuscript technically sound, and do the data support the conclusions?

Reviewer #1: Partly

Reviewer #2: Yes

3. Has the statistical analysis been performed appropriately and rigorously? 

Reviewer #1: No

Reviewer #2: Yes

4. Have the authors made all data underlying the findings in their manuscript fully available?

Reviewer #1: Yes

Reviewer #2: Yes

5. Is the manuscript presented in an intelligible fashion and written in standard English?

Reviewer #1: Yes

Reviewer #2: Yes

6. Review Comments to the Author

Reviewer #1: Thank you once again for inviting me to review further this manuscript. It is clear that the study is interesting and has some novelty. However, I think the authors need strong support for the statistical analysis part. With my previous comments, I was trying to encourage them to analyze their data in a robust way using LMM with the inclusion of random factors using R program instead of SPSS. The authors are stating that they used SPSS to do so and they even took out the R now from their data analysis section with this version. Still how they used the different statistical tools for the analysis is crude and not explicitly described. If the authors are willing to improve their manuscript by including these comments, I think the manuscript may be accepted for the publication without further review process.

Reviewer #2: You corrected all the comments and suggestions that I gave you in the second round. Hence, I suggest to the editor that it be published in the journal.

Good luck.

7. PLOS authors have the option to publish the peer review history of their article (what does this mean?). If published, this will include your full peer review and any attached files.

Reviewer #1: **Yes: **Debissa Lemessa

Reviewer #2: No

---

## [Author Response · Author response to Decision Letter 2]

5 Aug 2024

Title: Patterns of Primates Crop Raiding and the impacts on incomes of Smallholders across Mosaic agricultural Landscape of Wolaita Zone, Southern Ethiopia 

Manuscript Number: PONE-D-23-43746-R3

Response to Reviewers

We are great full to the editors and reviewers for their insightful and valuable comments on our paper. We have carefully considered the comments and tried our best to address every one of them. We hope the manuscript after careful revisions meet your higher standard journal. The authors welcome further constructive comments if any. We provided the point by point responses. All modifications in the manuscript have been highlighted in yellow color.

Sincerely,

Yigrem Deneke

Journal Requirements:

 Response

 The following reference was removed from the manuscript in the first-round revision (R1):

 The linear regression model results are reported as R2 values (proportion of variance accounted for), beta values (contribution to the model), t statistics (statistical significance of the contribution), and regression equations (the combination of variables best accounting for observed outcomes) (Sokal RR & Rohlf FJ, 1995). This was removed from the data analysis because the linear model only analyzes data with fixed factors. In contrast, the linear mixed model analyzes data with both fixed and random factors, along with the response variables.

The following nine references were added to the manuscript in the second-round revision (R2): 

 Hill, CM, Webber, AD, 2010; Hockings, KJ, Sousa, C, 2013; Freed, BZ, 2012; McGuinness, S, Taylor, D, 2014; Findlay, LJ, 2016; Gillingham, S, Lee, PC, 2003; Aharikundira, M, Tweheyo, M, 2011; Ango, TG, Börjeson, L, Senbeta, F, Hylander, K, 2014; Fang L, Hong, Y, Zhou Z, Chen, W, 2021. This is added to the manuscript because it includes important additional background information for this study.

The following reference was removed from the manuscript in the second & third-round revision (R2 &R3): 

 R core team. R, 2020. This reference was removed because the data were only analyzed using SPSS in the manuscript.

The following reference was added to the manuscript in the third round revision (R4):

 R core team. R, 2024. I used the latest version of the R program because it is robust for analyzing various datasets. This reference is now included to highlight the effectiveness and reliability of R in handling the spatio-temporal datasets.

I have replaced Table 5 and text, Figure 5, and Figure 6, which were analyzed using SPSS, with new versions that can be analyzed using the R program.

I have included the number of field experts and local farmers participated in this study

I have included a relevant remarks in the conclusion section

I have included the following supporting documents in the manuscript (R3):

S2 file. The rate of maize damage by olive baboons in different crop phonological stages was analyzed in both protected and open/control fields using R code.

S3 file. The rate of maize damage by grivet monkeys in different crop phenological stages was analyzed in both protected and open/control fields using R code.

S4 table 2. A linear mixed model of the maize damage rate by primates, considering different spatio-temporal variables, was analyzed using R code.

Response to Reviewer 1

Reviewer #1: Thank you once again for inviting me to review furthers this manuscript. It is clear that the study is interesting and has some novelty. However, I think the authors need strong support for the statistical analysis part. With my previous comments, I was trying to encourage them to analyze their data in a robust way using LMM with the inclusion of random factors using R program instead of SPSS. The authors are stating that they used SPSS to do so and they even took out the R now from their data analysis section with this version. Still how they used the different statistical tools for the analysis is crude and not explicitly described. If the authors are willing to improve their manuscript by including these comments, I think the manuscript may be accepted for the publication without further review process.

Response: Thank you very much for the comments. I included the analysis done with R, running the analysis using LMM to account for both fixed and random factors and response variables. This analysis is clearly indicated in the supporting documents. The different statistical tools used for the analysis are also explicitly described in the data analysis section.

Reviewer #2: You corrected all the comments and suggestions that I gave you in the second round. Hence, I suggest to the editor that it be published in the journal.

Good luck.

Response: Thank you very much.

---

## [Decision Letter · Decision Letter 3]

11 Sep 2024

PONE-D-23-43746R3Patterns of Primates Crop Raiding and the impacts on incomes of Smallholders across Mosaic agricultural Landscape of Wolaita Zone, Southern EthiopiaPLOS ONE

Dear Dr. Deneke,

Thank you for submitting your manuscript to PLOS ONE. After careful consideration, we feel that it has merit but does not fully meet PLOS ONE’s publication criteria as it currently stands. Therefore, we invite you to submit a revised version of the manuscript that addresses the points raised during the review process.

The first reviewer recommended the article for acceptance and the second one had concerns about the quality of what is presented as the background to the study within the introduction section. I agree with this and urge that the authors make a solid case for where there study fits within the body of knowledge existing on human-primate conflicts within the framework of crop raiding, not only in Ethiopia or Africa but actually throughout the tropical region. In short, a more thorough literature review, preferably with citations of research conducted within the past 1-6 years.

The third reviewer had even more serious concerns regarding the analytical and conceptual framework of the study, specifically feeling that some analyses were largely still unsound, many of them repetitive; that there was little justification for the large array of analytical tools employed by the authors. For instance could the authors try and reduce the number of statistical tools or more clearly explain how the various analyses do not duplicate each other? In addition, state what data model distribution  and link functions were applied to the LMMs

In addition:

Please ensure that towards the end of the abstract, you indicate the significant potential application of your findingsTo the declaration section on ethical statement, please add/confirm explicitly that: “In addition, there was no direct interaction between field personnel and the subjects (the primates) in such a way as to harm the animals or interfere with their freedom in nature such as by way of capture or trapping”Several subsections within Results and Discussion sections could be consolidated and merged so as to yield a maximum of no more than 4 subsections in each of them. This will make for easier readingPlease try and pay much more close attention to the quality or writing in terms of language, grammar, spelling and logical flow within paragraphs and the interconnections amongst paragraphs and sections. Seek the services of a professional English language EditorThe formatting of the reference list and figure as well as table captions and legends must strictly follow the guidelines of PLOSONE. If not, this alone will lead to rejection of the paper, which would be unfortunate, given the considerable time and effort the authors already invested in revising the manuscript so farConsolidate and minimize the number of illustrations: Aim for a maximum of 5 tables and 5 figures, and provide information regarding the extra illustrations within the text or leave them out altogetherThe revised manuscript should not bear any markings or color-highlighted sectionsTake note that the reviewers have uploaded PDFs of the manuscript in which they have indicated additional specific comments, and that you need to look at these and address all such issues in your revision, or provide sufficient rebuttals to the ones which you may be in disagreementMake sure that all the co-authors have thoroughly read and helped with the revision, and approved the final version before it is resubmitted

Carefully addressing these minor/major issues, as well as all the review comments will go a long way in making the paper more acceptable to the readership of PLOSONE

We look forward to receiving your revised manuscript.

Kind regards,

Nickson E. Otieno

Academic Editor

PLOS ONE

Journal Requirements:

Reviewers' comments:

Reviewer's Responses to Questions

**Comments to the Author**

1. If the authors have adequately addressed your comments raised in a previous round of review and you feel that this manuscript is now acceptable for publication, you may indicate that here to bypass the “Comments to the Author” section, enter your conflict of interest statement in the “Confidential to Editor” section, and submit your "Accept" recommendation.

Reviewer #3: (No Response)

Reviewer #4: (No Response)

Reviewer #5: All comments have been addressed

2. Is the manuscript technically sound, and do the data support the conclusions?

Reviewer #3: Partly

Reviewer #4: Yes

Reviewer #5: No

3. Has the statistical analysis been performed appropriately and rigorously? 

Reviewer #3: Yes

Reviewer #4: Yes

Reviewer #5: No

4. Have the authors made all data underlying the findings in their manuscript fully available?

Reviewer #3: Yes

Reviewer #4: Yes

Reviewer #5: Yes

5. Is the manuscript presented in an intelligible fashion and written in standard English?

Reviewer #3: No

Reviewer #4: Yes

Reviewer #5: Yes

6. Review Comments to the Author

Reviewer #3: While the manuscript is generally well-written, there are a few concerns. The introduction requires structural revisions for improved clarity and coherence. It should begin with a broader discussion of global human-wildlife conflict, particularly focusing on primates' effects on agriculture, to better establish the study's relevance. Additionally, the sections on the Sodo Zuriya and Damot Gale areas need to be integrated more effectively, highlighting their ecological significance and the importance of these regions for studying human-wildlife conflict. The introduction should also better articulate the problem statement, linking it to existing literature and identifying the research gap this study addresses. Methodologies such as camera traps and community-based studies should be reserved for the methods section. At the same time, the introduction should succinctly outline the study's objectives and emphasize the importance of the findings for conservation and conflict management. Improving the flow and ensuring a logical progression from general to specific information will enhance the introduction's impact. Another concern is the annotation of statistical results and terminology, which needs to be addressed for greater accuracy and clarity.

In conclusion, the study provides valuable insights into primates' foraging behaviour and maize damage across 25 small maize fields, both protected and non-protected. Overall, I was inclined to agree with the idea that this research is crucial for understanding the impacts on maize crops and developing strategies to mitigate wildlife conflict and support the livelihoods of smallholder farmers in the region. Additional comments are detailed in the attached PDF. Should these be addressed, I recommend the manuscript be accepted for publication.

Reviewer #4: (No Response)

Reviewer #5: This manuscript describes the results of a study of patterns of crop foraging by primate species and the impacts on the incomes of smallholders across the mosaic agricultural Landscape of Wolaita Zone.in Ethiopia. I recognize the hard work it has taken to conduct the fieldwork presented in this paper and the time authors have put into analyzing their data and writing the manuscript. However, the manuscript has several conceptual and analytical problems. The introduction lacks conceptual ideas. It should be seriously rewritten on the current Knowledge of crop foraging by wild animals worldwide. Therefore, the paper needs major revision to be accepted by PlOS ONE. It should be written based on the guidelines of PlOS ONE.

I included all my comments within the PDF file. The authors should use this file to correct those comments. I stated some crucial comments here.

1. The authors use a lot of statistical packages. I advise the authors to consult a statistician about their data for better analysis.

2. Some of the statistical values are not correct and clear too.

3. Why did the authors use two different kinds of statistical software? SPSS and R

4. Once the authors abbreviate phrases, they should use that abbreviation afterward. It is unnecessary to abbreviate here and there.

5. The author should say crop foraging instead of crop raiding as well as forage instead of raid throughout the paper

6. The authors should use a similar common name for baboons; olive baboons instead of anubis baboons

7. The method lacks clarity

8. I am so unhappy with the citation system. The authors seem no experience or they are careless. For example, a study conducted by, according to a paper by, as stated in reference

9. There are many tables and figures. Some of them are unnecessary. Please trim your study as short and understandable as possible.

7. PLOS authors have the option to publish the peer review history of their article (what does this mean?). If published, this will include your full peer review and any attached files.

Reviewer #3: No

Reviewer #4: No

Reviewer #5: No

---

## [Author Response · Author response to Decision Letter 3]

23 Sep 2024

Title: Patterns of Primates Crop Foraging and the impacts on incomes of Smallholders across Mosaic Agricultural Landscape of Wolaita Zone, Southern Ethiopia 

Manuscript Number: PONE-D-23-43746-R4

Response to Reviewers

We are great full to the editors and reviewers for their insightful and valuable comments on our paper. We have carefully considered the comments and tried our best to address every one of them. We hope the manuscript after careful revisions meet your higher standard journal. The authors welcome further constructive comments if any. We provided the point by point responses. All modifications in the manuscript have been highlighted in yellow color.

Sincerely,

Yigrem Deneke

Journal Requirements:

 Response

 The following four references were added to the manuscript in the fourth-round revision (R4): 

 Cuesta Hermira, AA, Michalski, F, 2022; Kifle, Z., Bekele, A, 2020; Kifle, Z., Bekele, A, 2021; Mesfin Matusal et al., 2023. This is added to the manuscript because it includes important additional background information for this study.

 The following reference was removed from the manuscript in the fourth-round revision (R4): 

 Hansen, LK, 2003. 

I have moved the following previously reported results to the supporting information section of the manuscript (R4):

S1 Fig. Location map of the study area (created with ESRI ArcGIS Desktop 10.8)

S2 Fig. Various prevention strategies (Wire mesh (A), Human guardian tower (B), Scarecrow (C), Thorny bush (D)) were assessed in eight experimental maize field sites to evaluate their effectiveness in deterring crop raiders. The study was conducted in maize field sites located in Gurumu Woide and Kokate Marachare (Photo credit: Yigrem Deneke).

S3 Fig. Diagrammatic example of a field map used by observers. HSE = house. GH = guard hut. SH = storage hut. Solid black lines = field boundary. Green objects = trees.

S4 Fig. The images above depict camera trap captures of various wildlife species observed in maize field sites located in Damota Mountain, Southern Ethiopia: (A) olive baboons (Papio anubis) (B) Grivet monkeys (Chlorocebus aethiops), (C) Porcupine (Hystrix cristata), and (D) Bushbuck (Tragelaphus scriptus) 

S5 Fig. Relative frequency of raid durations by primate CREs (n = 95).

S6 Fig. Relative frequency of raiding by primate CREs (n = 95)

S7 Fig. The number of baboon and grivet monkey field visits that did and did not involve crop-raiding events (CRE) on maize fields in the Highlands of Damota Mountain, April to September 2020 and 2021 years (n=367)

S8 Fig. The number of baboon and grivet monkey field visits that involved single- and multi-crop raiding events on maize fields in the Highlands of Damota Mountain, April to September 2020 and 2021 years (n=189).

S1 Table. Maize field and study plot size on the protective and non-protective maize fields

S2 Table. Farmer observation and reported of maize damage assessments (580 maize stem expected per plot except field no. 25 (see the text)

Response to Reviewer #1 

Abstract: The author stated that “…..while guarding is assumed to be an efficient protective strategy…when not implemented continuously.” What does it mean?? I don’t think it is scientific as such. You should be clear with the definition of guarding? Unless it is continuous keeping of crop fields in accordance with the animals’ activity time, it will not be guarding. So, rephrase this statement??? 

Response: It has been revised and properly rephrased.

Keywords/Phrases: Most journals require keywords to be listed alphabetically and they should be chosen from the abstract itself and should not be identical to words used in the title.

Response: It has been corrected and listed alphabetically.

Introduction: Page 3 Paragraph 2: the author hypothesized that ….frequency of crop raiding by large wild primates …..crop damage decrease with the distance from the forest edges ..” The hypothesis assumes a linear or predictable relationship between distance from the forest edge and primate behavior. Although the hypothesis presents a reasonable starting point, it simplifies complex ecological and behavioral relationships. For instance, primates might not always follow such a pattern due to factors like availability of food in the forest, group dynamics, and habituation to human activity, the size of the crop field, types of crops, seasonal variations, and weather conditions could also impact raiding behavior. However, the authors did not mention these factors in the hypothesis and overemphasized on distance.

Response: It has been revised. 

Methods

Page 5 Paragraph 3: The authors stated that they established 10 maize plots within 50 meters of the forest, while only 15 plots were set up at three different distance ranges: 100 meters, 200 meters, and 300 meters (with 5 plots per range). This means that the plots closest to the forest (at 50 meters) were twice of the number of plots to those in the subsequent distance ranges (100 meters, 200 meters, and 300 meters). Consequently, it is possible that a higher incidence of crop raiding might be observed on these more number of plots closest to the forest. This uneven distribution of plots could potentially lead to biased conclusions. Thus, caution should be taken. 

Page 5 Paragraph 4: The authors stated that “... we planted the high-yielding maize variety …” Did authors engage in planting maize seeds to the study plots???

Response: It has been revised. We compared maize damage assessments at varying distances by evaluating an open maize field located 50 meters from the forest edge, along with individual fields situated 100 meters, 200 meters, and 300 meters away from the forest edge.

Data analysis

Is there any scientific reason to use the old version of the SPSS software (16)??

Response: We used SPSS Version 16 because the results are consistent with those from more recent versions. The core statistical algorithms have remained largely unchanged across different releases. Key statistical computations in SPSS, such as the Chi-square test, Mann-Whitney U test, Spearman’s Rank Correlation Coefficient, t-test, one-way ANOVA, and F-test, have not significantly differed between versions. This means that when using SPSS Version 16 for standard statistical analyses, the results should match those obtained from more recent versions like SPSS 28, provided the same data and settings are used. The underlying statistical algorithms for these common tests have remained stable, so outputs such as p-values, confidence intervals, and coefficients will be consistent across versions under the same conditions

Results

The result section is staffed with large sized tables and too many figure. I recommend revising the larger tables (Table 1 and 3) or consider them for supplementary attachments.

Response: We used 5 tables and figures in the Results section; the remaining Table 1 and Table 3 have been moved to the supporting information.

Discussion

This section is well presented.

Response: Thank you very much

Conclusion

The conclusion is a short remarking of the key findings where the authors provided a concise overview of the main findings.

Response: Ok

Response to Reviewer #2 

Introduction 

The introduction lacks conceptual ideas. It should be seriously rewritten on the current Knowledge of crop foraging by wild animals worldwide. 

Response: It has been revised based on the current research on crop foraging by primates worldwide

I included all my comments within the PDF file. The authors should use this file to correct those comments. I stated some crucial comments here. 

1. The authors use a lot of statistical packages. I advise the authors to consult a statistician about their data for better analysis.

2. Some of the statistical values are not correct and clear too.

Response: Some of the statistical values have been revised and refined all the analysis throughout the manuscript. We consult a statistician for some analysis.

3. Why did the authors use two different kinds of statistical software? SPSS and R

Response: We use both SPSS and R for our analyses. SPSS is ideal for users who prefer a point-and-click interface and need to perform standard statistical analyses without learning to code. It is well-suited for researchers conducting routine analyses that do not require extensive customization. On the other hand, R is especially useful for handling complex data analysis tasks, such as spatio-temporal ecological datasets. It supports advanced statistical techniques (e.g., linear mixed models with random and fixed effects) and offers highly customizable visualizations (e.g., box plots, bar charts), providing far greater flexibility than SPSS.

4. Once the authors abbreviate phrases, they should use that abbreviation afterward. It is unnecessary to abbreviate here and there.

Response: It has been revised. 

5. The author should say crop foraging instead of crop raiding as well as forage instead of raid throughout the paper

Response: It has been revised and corrected throughout the manuscript.

6. The authors should use a similar common name for baboons; olive baboons instead of anubis baboons

Response: It has been revised to consistently use 'olive baboons' throughout the manuscript.

7. The method lacks clarity

Response: It has been revised, and this section is now clearly written.

8. I am so unhappy with the citation system. The authors seem no experience or they are careless. For example, a study conducted by, according to a paper by, as stated in reference

Response: It has been revised throughout the manuscript.

9. There are many tables and figures. Some of them are unnecessary. Please trim your study as short and understandable as possible.

Response: It has been revised. We used 5 tables and 5 figures in the Results section 

In addition:

Please ensure that towards the end of the abstract, you indicate the significant potential application of your findings

Response: The significant potential of this study has been included in the manuscript.

To the declaration section on ethical statement, please add/confirm explicitly that: “In addition, there was no direct interaction between field personnel and the subjects (the primates) in such a way as to harm the animals or interfere with their freedom in nature such as by way of capture or trapping”

Response: It has been incorporated in Ethics statement of the manuscript.

Several subsections within Results and Discussion sections could be consolidated and merged so as to yield a maximum of no more than 4 subsections in each of them. This will make for easier reading

Response: All the Results and Discussion sections have been consolidated, merging several headings into four main subsections.

Please try and pay much more close attention to the quality or writing in terms of language, grammar, spelling and logical flow within paragraphs and the interconnections amongst paragraphs and sections. Seek the services of a professional English language Editor.

Response: We have improved the language, grammar, spelling, and logical flow within paragraphs and sections throughout the manuscript.

The formatting of the reference list and figure as well as table captions and legends must strictly follow the guidelines of PLOSONE. If not, this alone will lead to rejection of the paper, which would be unfortunate, given the considerable time and effort the authors already invested in revising the manuscript so far

Response: It has been revised, and the manuscript has been written according to the PLOS ONE guidelines.

Consolidate and minimize the number of illustrations: Aim for a maximum of 5 tables and 5 figures, and provide information regarding the extra illustrations within the text or leave them out altogether

Response: We used 5 tables and 5 figures in the Results section, with the remaining tables and figures included in the supporting information.

Take note that the reviewers have uploaded PDFs of the manuscript in which they have indicated additional specific comments, and that you need to look at these and address all such issues in your revision.

Response: We have addressed and carefully corrected both the tracked changes and comments in the PDF made by the reviewers, providing point-by-point responses to each comment.

Finally, we would like to thank all the reviewers and editors for their comments and questions, which have greatly contributed to improving the quality of the paper.

---

## [Editor Report · Decision Letter 4]

29 Sep 2024

PONE-D-23-43746R4Patterns of Primates Crop Foraging and the impacts on incomes of Smallholders across Mosaic Agricultural Landscape of Wolaita Zone, Southern EthiopiaPLOS ONE

Dear Dr. Deneke,

Thank you for submitting your manuscript to PLOS ONE. After careful consideration, we feel that it has merit but does not fully meet PLOS ONE’s publication criteria as it currently stands. Therefore, we invite you to submit a revised version of the manuscript that addresses the points raised during the review process.

We look forward to receiving your revised manuscript.

Kind regards,

Nickson E. Otieno

Academic Editor

PLOS ONE

Journal Requirements:

**Additional Editor Comments:**

Abstract

Needs a brief statement of the study’s objectives, and a brief outline of the general methods used to collect information. Please ensure that in the first part of the abstract, there is something (1-2 sentences)outlining how the study was conducted, before mentioning the findings

Introduction

Needs to be expanded with more literature review of recent studies on crop damage by primates in the tropics generally but in Africa in particular (especially maize). Ensure you mention how your study fills in any gap left by such studies i.e, what is new about your findings. This was clearly emphasized by Reviewer #1 but the authors have not addressed it. Many studies related to this have been conducted in many parts of the tropics. The introduction part needs to be at least 2 pages longNeeds a statement towards the end, of hypotheses to justify linear modeling. Mention at least one hypothesis and what you expected to find out from the sudyRemove the statement in which insects are mentioned as crop raiders. This study is bout primates

Tables and figures

Remove vertical lines in tablesWithin tables, center all texts and numbers within columns and rows except column titlesStandardize font types and size in all tables according to journal guidelinesTable legends and figure captions must contain enough details to be able to stand aloneIf abbreviations are used in figure axes such as in Fig 5, they must be fully defined in the captions text

**In general**

The language is still below par, and needs professional editing for English. (There are very many affordable but effective ones available ones which can be contacted online to complete editing within 48 hours). Therefore, in the revised edition of the MS, a professional English proof reading certificate must be submitted.Only the first word of each title or subtitle should be capitalizedFor decimal numbers, standardize  and round off to a maximum of 3 decimal pointsThroughout the manuscript, adopt a third-party  stance in presentation of the write-up  e’g “the study was conducted …” instead of “We conducted the study …”;  and “The experimental set up … “ instead of “Our experimental set up …”Do not rush to resubmit the revised manuscript, if these issues are nit addressed. You have until te end of October 2024.

In summary, this is a good opportunity for the authors to rescue the paper after the previous efforts at revision, but still, if any of the above concerns and issues are not addressed in full, the revised paper will be rejected outright.

---

## [Author Response · Author response to Decision Letter 4]

25 Oct 2024

Title: Patterns of Primates Crop Foraging and the Impacts on Incomes of Smallholders across the Mosaic Agricultural Landscape of Wolaita Zone, Southern Ethiopia

Manuscript Number: PONE-D-23-43746-R5

Responses to reviewers

We are grateful to the editors and reviewers for their insightful and valuable comments on our paper. We have carefully considered the comments and tried our best to address every one of them. We hope the manuscript after careful revisions meets your higher standard journal. The authors welcome further constructive comments if any. We provided the point-by- point responses. All modifications in the manuscript have been highlighted in yellow.

Sincerely,

Yigrem Deneke

Journal Requirements:

 Response

 The following four references were added to the manuscript in the fourth-round revision (R4): 

 Naughton-Treves L, Treves A. 2005

 Alemu M, Berihun DC, Lokossou J, Yismaw, B. 2024 

 Abate T, Shiferaw B, Menkir A, Wegary D, Kebede Y, Tesfaye K, Keno T. 2015

 Prasanna BM, Palacios-Rojas N, Hossain F, Muthusamy V, Menkir A, Dhliwayo T, Ndhlela T, San Vicente FM, Nair SK, Vivek BS, Zhang X, Olsen M, Fan, X. 2020

 Ikhuluru WE, Imboma ME, Liseche SE, Milemele MJ, Shilabiga SD, Cords M. 2023

 Koirala S, Garber PA, Somasundaram D, Katuwal HB, Ren B, Huang C, Li M. 2021

 Hill CM. 2020

 Wiafe ED. 2019

 Mwakatobe A, Nyahongo J, Janemary NJ, Røskaft E. 2014

 Jaleta M, Tekalign W. 2023

 Masha M, Yirgu T, Debele M, Belayneh M. 2021

 Abendroth LJ, Elmore RW, Boyer MJ, Marlay SK. 2011

 Wolaita District Agricultural Office (WDAO). 2021. This is added to the manuscript because it includes important additional background information for this study.

Responses to reviewers 

Abstract

• It needs a brief statement of the study’s objectives and a brief outline of the general methods used to collect information. Please ensure that in the first part of the abstract, there is something (1-2 sentences)outlining how the study was conducted, before mentioning the findings

Response: It has been revised accordingly 

Introduction

• It needs to be expanded with more literature review of recent studies on crop damage by primates in the tropics generally but in Africa in particular (especially maize). Ensure you mention how your study fills in any gap left by such studies i.e, what is new about your findings. This was clearly emphasized by Reviewer #1 but the authors have not addressed it. Many studies related to this have been conducted in many parts of the tropics. The introduction part needs to be at least 2 pages long

• It needs a statement towards the end, of hypotheses to justify linear modeling. Mention at least one hypothesis and what you expected to find out from the study

• Remove the statement in which insects are mentioned as crop raiders. This study is bout primates

Response: The manuscript has been revised to include an expanded literature review on recent studies of crop damage caused by primates in tropical regions, with a particular focus on Africa and maize. We have addressed the gap in previous research and highlighted the novel aspects of our findings. Additionally, we incorporated a hypothesis involving linear mixed modeling and removed references to insect-related crop damage.

Tables and figures

• Remove vertical lines in tables

• Within tables, center all texts and numbers within columns and rows except column titles

• Standardize font types and size in all tables according to journal guidelines

• Table legends and figure captions must contain enough details to be able to stand alone

• If abbreviations are used in figure axes such as in Fig 5, they must be fully defined in the captions text

Response: The manuscript has been revised to ensure that the font types and sizes in all tables adhere to the PLOS ONE guidelines. The table legends and figure captions have been modified to stand alone, and the caption text for Figure 5 has also been corrected.

In general

• The language is still below par, and needs professional editing for English. (There are very many affordable but effective ones available ones which can be contacted online to complete editing within 48 hours). Therefore, in the revised edition of the MS, a professional English proof reading certificate must be submitted.

• Only the first word of each title or subtitle should be capitalized

• For decimal numbers, standardize and round off to a maximum of 3 decimal points

• Throughout the manuscript, adopt a third-party stance in presentation of the write-up e’g “the study was conducted …” instead of “We conducted the study …”; and “The experimental set up … “ instead of “Our experimental set up …”

• Do not rush to resubmit the revised manuscript, if these issues are nit addressed. You have until to end of October 2024.

Response: The manuscript has been revised with professional English editing, and the editor certificate is included as supporting information. All other issues have been thoroughly addressed, resulting in significant improvements to the manuscript.

Comments about the PDF and word documents

Question # 1. Is there any difference between CRE and CFE? If not why didn’t take one of them.

Response: This is a valid point raised by the reviewer. The two terms differ in context: Actual crop feeding events (CFEs) refer to instances where primates actively enter a field and feed on crops, including observations of animals consuming leaves, fruits, seeds, and other plant parts. In contrast, potential crop raiding events (CREs) refer to situations where primates enter or approach a field but do not necessarily feed on the crops. In these cases, the animals may trample the field, cause stem or root damage, investigate, or simply pass through without consuming any crops. By categorizing both actual CFEs and potential CREs under the umbrella of crop foraging events, we can explore the overall interaction of primates with agricultural systems, capturing both active feeding behaviors and exploratory actions that may lead to crop damage.

Question # 2. Your respondents are too small in number. Why? Are they enough for this study

Response: Our experimental study involved 25 maize plots, with one farmer assigned to each plot (a total of 25 farmers) to conduct field observations and record maize damage caused by primates. This arrangement effectively minimized disturbances by primates and allowed for more accurate assessments of primate-related maize damage. The number of farmers was adequate, as the primates had easy access to the fields. The selected farmers, who lived in the surrounding Damota community-managed areas, were long-time residents with extensive experience in managing human-primate conflicts, particularly related to crop damage. As members of the Damota Forest Protection group, they provided valuable insights, enabling us to gather comprehensive information on the patterns and extent of crop foraging events by primates.

All the other comments in the track changes (PDF) and the Word documents have been thoroughly revised and addressed in the manuscript.

Finally, we would like to thank all the reviewers and editors for their comments and questions, which have greatly contributed to improving the quality of the paper.

---

## [Editor Report · Decision Letter 5]

1 Nov 2024

Patterns of Primates Crop Foraging and the impacts on incomes of Smallholders across Mosaic Agricultural Landscape of Wolaita Zone, Southern Ethiopia

PONE-D-23-43746R5

Dear Dr. Deneke,

We’re pleased to inform you that your manuscript has been judged scientifically suitable for publication and will be formally accepted for publication once it meets all outstanding technical requirements.

Kind regards,

Nickson E. Otieno

Academic Editor

PLOS ONE

Additional Editor Comments (optional):

Before the article can be considered for acceptance towards publication, the authors must  either address these few additional outstanding concerns that the reviewers had, and which the authors did not mention in their response document to the reviewers’ comments or in the revised article. Or provide good solid reasons as to why they think those issues are not worthy of being addressed/ Please answer them point by point (i.e from point 1 to point 4, NOT A ONE GENERAL PARAGRAPH ANSWER):

*The authors compared the extent of crop damage by primate species in protected and non-protected maize fields. These two types of fields should have been found at an equal distance from the forest edge of the primate habitat. If they were found at different distances, how can you be sure to answer your research questions? I.e., Okay you can measure the extent of crop damage by considering distances. But to measure the extent of crop damage between protected and unprotected fields, the two types of crop fields should be located at equal distances from the forest edges.  *

*In all distances 50m, 100m, 200m, and 300m each protected fields method (wire mesh, human guardians, scarecrows, and thorny bushy) for maize fields should be available. If not your experiment did not identify the better protection method and provide mitigation measures. You can evaluate those protection methods.  *

*The authors set up 15 non-protected maize fields and 10 protected maize fields. Why did you set up different numbers of maize fields? You should set up equal numbers for both protected and non-protected maize fields.*

*Why did you guard the maize field only for seven days? Maize took at least three months to harvest?. *

In addition,

Please strictly follow, adhere to and effect the following points in improving the structuring  and outlay of the article:

The article text must be double-spaced and use the same font type throughout. At the moment, this is not the case in your paperEvery page must have “continuous  line-numbering”, including pages with tables figures and supplementary materialTest must be formatted to leave a margin of at least 1.5 inches on both sidesEnsure that all co-authors, especially the one listed last (H. L.) read the manuscript and help with final corrections in language and structure before it is resubmitted to PLOS ONE. It appears they were not contacted by you in the last round of revisionIn adherence to journal guidelines, Avoid capitalizing first letters of worlds in the titles and subtitles unless they are names of places of common names of animalsRemove all text in italics unless used for animal scientific namesDefine GPS in full when fist mentioned. Do this for all other abbreviations in the manuscriptAvoid using italic font in the text except for animal scientific names of local-dialect wordsFormat the article to fit in the page and leave at least 1.5 inch margins. Some tables have been seen to be overlapping pages. Seek support from someone  experienced In document layout formattingTables must not have an vertical lines

Please look at the article, which has been attached for you, and which bears the specific comments from the editor (some of which overlap with the ones outlined above).
---

## [Editor Report · Acceptance letter]

7 Nov 2024

PONE-D-23-43746R5 

PLOS ONE

Dear Dr. Deneke, 

I'm pleased to inform you that your manuscript has been deemed suitable for publication in PLOS ONE. Congratulations! Your manuscript is now being handed over to our production team.

Kind regards, 

on behalf of

Dr. Nickson E. Otieno 

Academic Editor

PLOS ONE